# A Promising Approach to Psoriasis Vulgaris Management with N-Acetylcysteine and Vitamin E: Targeting the Interplay of Inflammatory and Oxidative Stress

**DOI:** 10.3390/biomedicines13061275

**Published:** 2025-05-22

**Authors:** Nira Elkalla, Manal H. Elhamammsy, Nermeen Ibrahim Bedair, Ola Elazazy, Amal A. El Kholy

**Affiliations:** 1Clinical Pharmacy Department, Faculty of Pharmacy, Badr University in Cairo (BUC), Cairo 11829, Egypt; 2Clinical Pharmacy, Clinical Pharmacy Department, Faculty of Pharmacy, Ain Shams University, Cairo 11566, Egypt; manal.elhamamsy@pharma.asu.edu.eg (M.H.E.); amal.elkhouly@pharma.asu.edu.eg (A.A.E.K.); 3Dermatology, Andrology, Sexual Medicine and STDs Department, Faculty of Medicine, Helwan University, Cairo 11795, Egypt; nermeen.bedair@med.helwan.edu.eg; 4Biochemistry Department, Faculty of Pharmacy, Badr University in Cairo (BUC), Cairo 11829, Egypt

**Keywords:** psoriasis, PASI, IL-36γ, MDA, N-acetylcysteine, vitamin E

## Abstract

**Background:** Psoriasis is a persistent, inflammatory skin disease with autoimmune characteristics. Beyond the obvious signs of skin lesions, it has negative systemic repercussions that impair the patient’s quality of life. This study aimed to determine the effectiveness of N-acetylcysteine (NAC) alone or in combination with Vitamin E in the treatment of mild to moderate active psoriasis vulgaris. **Methods:** This study was an open-label, prospective, randomized, controlled interventional clinical trial conducted at Cairo Hospital for Dermatology and Venereology (Al-Haud Al-Marsoud). In total, 45 patients with mild to moderate symptoms were randomly assigned to three groups, with fifteen patients each, as follows: the control group received the standard psoriatic treatment of topical steroids and salicylic acid; the acetylcysteine group received standard psoriatic treatment in addition to NAC 600 mg per day 30 min prior to breakfast for 8 weeks; and the acetylcysteine and Vitamin E group received standard psoriatic treatment in addition to NAC 600 mg per day, in a similar way of dosing like the previous group, and Vitamin E 1000 mg per day. All participants performed a comprehensive assessment including hematological parameters, the Psoriasis Area and Severity Index (PASI), the Dermatology Life Quality Index (DLQI), malondialdehyde (MDA), and interleukin-36 gamma (IL-36γ). **Results:** The treatment strategy involving the use of NAC alone and in combination with Vitamin E showed significant improvement in the assessed parameters compared to the control group receiving conventional therapy. The acetylcysteine group showed improvements of 41% in PASI and 49.4% in DLQI, a decrease of 34.3% in MDA, and a decrease of 31% in IL-36γ. Similarly, the acetylcysteine and Vitamin E group showed improvements of 52% in PASI and 42% in DLQI, a decrease of 37% in MDA, and a decrease of 35% in IL-36γ. There were no significant differences found between the N-acetylcysteine and N-acetylcysteine and Vitamin E groups. Moreover, significant positive correlations were found between MDA, IL-36γ, and PASI at baseline and after the third follow-up. **Conclusions:** This study found promising therapeutic benefits in the addition of NAC to the conventional therapy in psoriatic patients with mild to moderate symptoms, as it significantly improved psoriasis disease outcomes and improved the patient’s quality of life. However, the addition of Vitamin E to the NAC regimen did not show additional benefits.

## 1. Introduction

Psoriasis is characterized by chronic inflammation of the skin. It affects two percent of the population worldwide, ranging to almost 125 million people. About 1–3 percent of the general population is thought to have plaque psoriasis in African and Middle Eastern nations, which is similar to what is seen in Western nations [1]. Psoriasis is a genetic, immune-mediated disorder that manifests itself in the skin, joints, or both. It is a form of hyperkeratosis characterized by acanthosis and parakeratosis due to the accretion of the cells [2].

Inflammation is a reaction that occurs in the body because of recognition of harm or a pathogen that attacks it. The damage activates the immune system, generating inflammatory mediators, including cytokines and metabolites of arachidonic acid such as leukotrienes and prostaglandins, including proteins like C-reactive protein (CRP). Uncontrolled acute inflammation results in continuous activation of the immune system, putting the body in a continuous alert state [3]. The term “autoimmune disease” refers to a condition in which the body occasionally targets its own healthy cells rather than the harmful pathogens that are invading the body. This entails making autoantibodies, which keep the body constantly inflamed. It has been noted that T-cell and B-cell overactivity are key factors in autoimmune disorders. This demonstrates the close relationship between autoimmune disorders and chronic inflammation [4].

Although the rationale for developing psoriasis is still unknown, some causes are evident, like an overactive immune system, genetic mutations, or environmental triggers. Continuous immune system activation and sustained inflammation cause an excessive number of free radicals to be produced and enter the body in a condition known as oxidative stress [5]. Psoriasis has a complex pathogenetic pathway, but one possible component is a decrease in the body’s antioxidant capability due to an enzyme deficiency or mutation, such as glutathione S-transferase (GST) M1/T1 [6]. When the natural antioxidant system is unable to sufficiently eliminate oxidative stress, these oxidized radicals generated and amplified post-tissue injury due to inflammation promote a destructive loop [7].

Psoriasis has seven different types classified according to how the skin looks and the character of the patch: vulgaris psoriasis—the most frequent, inverse psoriasis, guttate psoriasis, pustular psoriasis, nail psoriasis, erythrodermic psoriasis, and psoriatic arthritis. Plaque psoriasis is highly prevalent, accounting for 90% of psoriasis patients. Plaque psoriasis is typically concentrated on the knees, elbows, or trunk and characterized by big, oval, circular lesions that are reddish or pink in color [8].

It is necessary to utilize specific techniques to grade the severity and extent of psoriasis, even though the clinical results for psoriasis patients are often evident. The Psoriasis Area and Severity Index (PASI) score worksheet rates the severity of psoriatic lesions according to the covered area. It is mostly done to monitor the severity of the condition both before and throughout therapy, which enables physicians to assess the efficacy of the patient’s treatment strategy [9]. Moreover, the Dermatology Life Quality Index (DLQI) is a brief questionnaire outlining how the illness and its treatment have affected the patient’s day-to-day activities. It is also recommended to use it in conjunction with PASI, as it is not specific for psoriasis, and physicians cannot rely on it alone to fully evaluate the condition [10]. Even though psoriasis patients are judged highly subjectively, treatment objectives are often explicit and objective. Therefore, the primary goal should be to lessen skin inflammation in all its forms, including discomfort, redness, and/or swelling, in addition to improving people’s quality of life [11].

Due to their severely reduced productivity, some patients experience significant psychological, social, and financial effects simply from having psoriasis. A PASI of three or less after three months of treatment would be considered satisfactory. The DLQI should also be in line with the PASI score, aiming to be three or less. A routine checkup every six months would be recommended. Adjustments should be made if the objective is not accomplished after three to four months of therapy [12].

To achieve the desired outcome and avoid problems, it is crucial to create a personalized, effective treatment plan for each patient based on their symptoms and comorbidities. The most prevalent complication is psoriatic arthritis (PsA), which affects 30% of psoriasis patients and causes stiff, painful joints, particularly in the fingers, knees, and elbows [13]. One concerning complication is the development of metabolic syndrome, a cluster of coexisting conditions that significantly increases the risk of cardiovascular diseases [14].

Unfortunately, there is no cure for psoriasis vulgaris; thus, therapeutic goals are to ultimately improve the quality of life, regulate the condition, and prevent complications. Topical treatments are the first category and are typically advised as a first line of treatment for mild to moderate patients. Steroids are the mainstay of therapy, and help lessen the itching that intensifies flare-ups [15].

Calcipotriol, a vitamin D analog, is another powerful topical treatment for psoriasis patients. It has anti-inflammatory properties and slows down the hyperproliferation of keratinocytes by inhibiting T lymphocyte activity, and alongside calcitriol, both orally and topically, these were effective options for treating psoriasis [16]. Additionally, tacrolimus and other calcineurin inhibitors are effective therapeutic choices, particularly for sensitive areas such as the face or genitalia. It lessens inflammation by lowering immune system activity [17,18]. Phototherapy is an additional treatment option, particularly for moderate to severe cases. In most instances, UVB phototherapy is the safest choice, while UVA is not advised unless necessary [19,20].

In moderate to severe cases with a PASI score greater than 10, systemic therapies including both non-biologic and biologic agents are commonly used. Methotrexate, cyclosporin, and Acitretin are non-biologic possibilities. The primary treatment option for persistent plaque psoriasis, particularly if psoriasis arthritis has been manifested, is methotrexate [21]. Cyclosporine, a calcineurin inhibitor that suppresses the immune system, is also used to treat plaque psoriasis. The third-line treatment is Acitretin, a daily oral retinoid used to reduce the development of new skin cells [22]. On the other hand, the use of biologics, such as Etanercept and Ustekinumab, is typically reserved for severe cases of psoriatic arthritis or when prior therapies have failed. These medications work by targeting multiple pathways involved in the pathophysiology of psoriasis, leading to significant improvements in disease stability and symptom relief [23].

Nonpharmacological therapy for psoriasis vulgaris is seen to be just as necessary and beneficial as pharmacological treatments. Keeping the body cool and following a well-balanced diet like the Mediterranean diet, which is high in fiber, fruits, vegetables, proteins, nuts, and extra virgin olive oil, are very beneficial. Supplementing the diet with fish oil omega-3 or oral vitamin D was found to have anti-inflammatory and healing properties for psoriasis patches [24,25]. Several conventional alternative methods may be beneficial for psoriasis patients, such as cupping treatment and hematophagous creatures like leeches [26,27].

N-acetylcysteine (NAC) is an amino acid that serves as the building block for L-cysteine. It works in two pathways. Firstly, it raises the body’s glutathione levels through an increase in glutathione S-transferase (GST) activity, which has a potent antioxidant impact. It is regarded as a strong scavenger of free radicals, particularly oxygen radicals [28]. Secondly, it reduces inflammatory cytokines such as interleukin (IL)-1β, IL-6, IL-36, and tumor necrosis factor-alpha (TNF-α), which contribute to its anti-inflammatory potential. As a result, it inhibits apoptosis and oxygen-related genotoxicity [29]. NAC is also highly beneficial for treating inflammatory conditions such as ulcerative colitis, asthma, and liver problems that are characterized by an elevated inflammatory response in the body [30]. These findings pave the way for studying its role in psoriasis.

Among the essential fat-soluble vitamins, Vitamin E offers numerous benefits for the human body. Its antioxidant activity is unique as it neutralizes free radicals when they arise and prevents their production from normal biological reactions such as energy production and fat metabolism [31]. Vitamin E also is integrated into biological membranes, thereby maintaining cell membrane integrity and protecting the cell against damage by inhibiting protein oxidation and lipid peroxidation [32]. Thus, it could prevent certain serious illnesses including cancer, diabetes, and dementia, which are linked to reactive oxygen species.

As psoriasis vulgaris is primarily a combination of inflammation and a high level of oxidation, it is believed that an effective and powerful antioxidant and anti-inflammatory agent is necessary to treat and/or prevent further tissue damage efficiently. Thus, the objective of this study was to explore the impact of incorporating NAC either alone or in conjunction with Vitamin E into conventional therapy for improving the clinical outcome, oxidative stress, and inflammation in patients with mild to moderate psoriasis vulgaris.

## 2. Materials and Methods

### 2.1. Design

This study was an open-label, prospective, randomized, controlled interventional clinical trial, to evaluate the effect of adding N-acetylcysteine alone or in combination with Vitamin E to conventional therapy in patients with mild or moderate psoriasis vulgaris.

### 2.2. Setting

This study was conducted at Cairo Hospital for Dermatology and Venereology (Al-Haud Al-Marsoud), Cairo, Egypt in the period between November 2022 and September 2023.

### 2.3. Ethical Consideration

The patients were informed of the study’s objectives, as well as the potential risks and benefits, by the principal investigator. A written informed consent document was completed by each patient. The Research Ethics Committee of Ain Shams University approved the study (ACUC-FP-ASU-RHDIRB20202110301 REC NO. 153, date: 20 December 2022), the Research Ethics Committee of Badr University in Cairo approved the study (BUC-IACUC-220911-3, date: 11 September December 2022), and the Research Ethics Committee of Training and research sector, Ministry of Health, Egypt, approved the study (COM.NO./DEC.NO.:18-2022/31, date: 2 November 2022). The research project was prospectively registered in the Protocol Registration and Results System (PRS) in Clinical Trials.gov (NCT05906498). The ethical standards of the Declaration of Helsinki, World Medical Association, 2013 were adhered to in all aspects of this study.

### 2.4. Study Subjects and Treatment

Patient enrollment was carried out in conformity with the inclusion and exclusion criteria established in the study design before its commencement.

The following were the inclusion criteria: male and female adult patients aged 18 to 65 years with mild to moderate psoriasis in the active phase (PASI ≤ 5 for mild psoriasis and 5–10 for moderate psoriasis) [33].

In addition, the criteria for patient exclusion were as follows: inactive psoriasis vulgaris; any other autoimmune disease; alcohol consumption; pregnancy or breastfeeding; any severe or systemic failure (e.g., cardiovascular, renal, or respiratory); history of chronic diseases, e.g., hypertension; history of bleeding, ulcers, or uncontrollable heartburn; any major psychiatric or mental illness; administration of any antioxidants in the previous three months; or administration of anticancer medications that can exacerbate psoriasis vulgaris, such as actinomycin, vinblastine, mercaptopurine, and radiation.

In total, 60 individuals with mild to moderate psoriasis vulgaris were screened for eligibility. Four individuals declined to participate, and six did not satisfy the inclusion/exclusion criteria. Five individuals were excluded prior to randomization due to one or more of the exclusion criteria, while 45 participants were enrolled. All eligible patients were assigned at random to one of the three groups, each consisting of 15 patients: the **control group**, where patients received the standard psoriatic treatment of topical steroids (Triamcinolone acetonide 0.5%, once daily before sleep), salicylic acid (2% once daily), and Urea (10% twice daily) [34]; the **acetylcysteine group,** where patients received standard psoriatic treatment in addition to acetylcysteine 600 mg effervescent granules (Acetylcistein^®^ 600 mg Effervescent instant Granules, South Egypt Drug Industries company (SEDICO), 6th of October City, Egypt) per day 30 min before breakfast for 8 weeks [35,36]; and the **acetylcysteine and Vitamin E group**, where patients received standard psoriatic treatment in addition to acetylcysteine 600 mg effervescent granules (Acetylcistein^®^ 600 mg Effervescent instant Granules, South Egypt Drug Industries company (SEDICO), 6th of October City, Egypt)) per day 30 min prior to breakfast for 8 weeks and one Vitamin E 1000 mg soft gelatin capsule per day (VITAMIN E^®^ 1000 mg soft gelatin capsules, Safe for Pharmaceuticals (Safe Pharma) Pharco Pharmaceuticals, Alexandria, Egypt) [37]. Follow-up assessments were conducted in parallel with the assigned course of treatment.

The allocation of subjects to the three study arms and study design is shown in the CONSORT diagram (Figure 1).


*Randomization Technique:*


Using computer-generated random numbers, eligible participants who met the inclusion criteria were randomly allocated into three groups. The distribution between the control group, the acetylcysteine group, and the acetylcysteine and Vitamin E group was maintained at a ratio of 1:1:1. It was unfeasible to blind investigators to the different treatment groups.

### 2.5. Baseline Assessment

According to the inclusion and exclusion criteria, the baseline demographic data for the selected patients were obtained, and then blood samples were withdrawn via venous phlebotomy and placed in a suitable vacutainer by a trained nurse. The collected blood samples were centrifuged for 10 min at 4000 rpm and the plasma was separated and stored at −80 °C for the subsequent analysis.

For all patients, the following data were collected: demographics and clinical feature assessments (age, sex, smoking status, family history, PASI, and DLQI); and a routine laboratory assessment which included the following: liver function tests (ALT, AST, and albumin), lipid profile (total cholesterol, triglycerides, LDL, and HDL), fasting blood glucose test (FBG), complete blood count (CBC), non-specific inflammatory markers (CRP and ESR), the oxidative stress marker malondialdehyde (MDA), and the inflammatory marker interleukin-36 (IL-36).

### 2.6. Outcome Evaluation and Follow-Up

Throughout a span of 3 months, patient follow-ups were conducted monthly with comprehensive assessments and examinations of the following aspects.

#### 2.6.1. Psoriasis Area and Severity Index (PASI) [33]

This was performed by the same evaluator to monitor the severity of the condition before and throughout therapy. The following formula was used to calculate the score:PASI = 0.1 (Eh + lh + Sh) Ah + 0.2 (Eu + lu + Su) Au + 0.3 (Et +lt + St) At + 0.4 (El + ll + Sl) Al
where E = erythema, I = induration, S = scaling, U = upper extremities, t = trunk, and L = lower extremities.

When the PASI score was less than 5, it was considered mild; between 5 and 10, it was considered moderate; and greater than 10, it was considered severe.

#### 2.6.2. Dermatology Life Quality Index (DLQI) [38]

Before the start of the study, and during the first visit, each patient answered the DLQI questionnaire by themselves, in their mother language. The scores were summed up and recorded. Then, at the last visit, the patient took the same questionnaire and answered it. The scores were summed up and recorded, to be compared with the records of the first visit. For illiterate patients, the questionnaire was asked in the form of voice questions, and the answers were recorded, and the scores were summed up.

#### 2.6.3. Liver Function Test (ALT, AST, and Albumin)

The commercial colorimetric assay kits purchased from Spectrum Diagnostics (Cairo, Egypt) were used to evaluate the liver injury biochemical markers, including aspartate aminotransferase (AST) (Cat No: 260001) and alanine aminotransferase (ALT) (Cat No: 264001), as per the manufacturer’s directions. Additionally, the chronicity marker albumin was detected using a single reagent colorimetric kit from DIALAB (Cat No: DT1001, Wiener Neudorf, Austria), according to the manufacturer’s directions.

#### 2.6.4. Lipid Profile (Total Cholesterol, Triglycerides, LDL, and HDL)

The commercial colorimetric assay kits purchased from Spectrum Diagnostics (Cairo, Egypt) were used to evaluate the total cholesterol (TC) (Cat No: 230001), triglycerides (TG) (Cat No: 314001), and high-density lipoprotein (HDL) (Cat No: 266001), as per the manufacturer’s instructions. Regarding HDL quantification, low-density lipoprotein (LDL) and very low-density lipoprotein (VLDL) were precipitated with phosphotungestate and magnesium ions followed by centrifugation, and the supernatant was used for HDL assessment utilizing the same technique used for TC. For the LDL determination, the Friedewald equation was used as follows:LDL-C = Total Cholesterol − HDL-C − (Triglycerides ÷ 5)

#### 2.6.5. Fasting Blood Glucose Test (FBG)

Glucose was determined colorimetrically utilizing a commercial assay kit from Spectrum Diagnostics (Cat No: 250001, Cairo, Egypt), following the manufacturer’s instructions.

#### 2.6.6. Complete Blood Count (CBC) (TLC Count, RBC Count, HB, HCT, Platelet Count, and RBC Indices)

After the blood sample was collected on heparin, it was sent to a laboratory with automated hematology analyzers to perform the analysis.

#### 2.6.7. Non-Specific Inflammatory Markers (CRP and ESR)

The C-reactive protein (CRP) concentration was determined utilizing a commercial solid-phase direct sandwich enzyme-linked immunosorbent assay procured from Sigma Aldrich (Cat No: SE120041, St. Louis, MO, USA), following the manufacturer’s instructions.

Regarding the erythrocyte sedimentation rate (ESR) estimation, the Westergren method was employed. The blood was drawn into a tube containing sodium citrate anticoagulant, diluted with sodium citrate in a 4:1 ratio in a Westergren tube; the tube was allowed to stand undisturbed for exactly one hour, and then the distance that the red blood cells sedimented from the top of the tube to the meniscus of the plasma was recorded.

#### 2.6.8. Oxidative Stress Marker Malondialdehyde (MDA)

Malondialdehyde (MDA) was analyzed using a colorimetric assay supplied by Biovision (Cat No: K739-100, Milpitas, CA, USA), following the manufacturer’s instructions.

#### 2.6.9. Inflammatory Marker Interleukin-36 Gamma (IL-36γ)

Interleukin-36γ (IL-36γ) was quantified via a sandwich enzyme-linked immune-sorbent assay procured from FineTest^®^ (Cat No: EH0628, Wuhan, China), following the manufacturer’s instructions.

In between visits, patients were contacted via phone to monitor any side effects, using a side effect reporting card. The primary outcomes were the clinical improvement of the inflammatory patches and the reduction in IL-36γ and MDA. Secondary outcome measures included the improvement in quality of life and prevention of complications of the disease like psoriatic arthritis, Parkinson’s disease, diabetes mellitus, and anxiety.

### 2.7. Statistical Analysis

All statistical analyses were performed using GraphPad Prism for Windows version 8.0 (ISI^®^ software, San Diego, CA, USA). For categorical variables, a chi-square test was used, while, for quantitative data, the normality was assessed via the Shapiro–Wilk test, and ANOVA was implemented to evaluate the significance of the outcomes between groups, followed by Tukey’s post-hoc test for normally distributed data. The Kruskal–Wallis Test was used for non-parametric data. Two-way ANOVA was used to evaluate the significance of the outcomes before and after treatment. The results were presented as means ± standard deviation (SD) or median (range). The significance threshold was considered at *p* < 0.05.

### 2.8. Sample Size Estimation

Owing to the lack of previous studies examining the effect of acetylcysteine and Vitamin E on active psoriasis vulgaris patients, we expected a large effect size of 0.5 on the PASI score. For a desired power of 0.80 and an alpha error of 0.05, a minimum sample size of 13 subjects per group (total 39 subjects) would be required. The sample would be increased by 15% to compensate for the loss to follow-up, to be 15 cases in each group (total 45 cases). The G*Power© program (Institut für Experimentelle Psychologie, Heinrich Heine Universität, Düsseldorf, Germany) version 3.1.9.2 was used to estimate the sample size.

## 3. Results

The study was designed to include 45 mild to moderate psoriatic patients selected from the Cairo Hospital for Dermatology and Venereology (Al-Haud Al-Marsoud), Cairo, Egypt in the period between November 2022 and September 2023. These patients were randomly assigned into three groups: the **control group**, which included 15 cases who received the standard psoriatic treatment of topical steroids and salicylic acid; the **acetylcysteine group**, which included 15 cases who received standard psoriatic treatment in addition to acetylcysteine 600 mg effervescent granules per day 30 min before breakfast for 8 weeks; and the **acetylcysteine and Vitamin E group**, which included 15 cases who received standard psoriatic treatment in addition to acetylcysteine 600 mg effervescent granules 30 min before breakfast and one Vitamin E 1000 mg soft gelatin capsule per day for 8 weeks.

### 3.1. Demographics and Clinical Characteristics of the Studied Groups

Regarding demographic data, the mean age in the control group was 43.1 ± 8.3, in the acetylcysteine group was 40.00 ± 12.65, and in the acetylcysteine and Vitamin E group was 41.93 ± 9.30, without any significant differences between groups (*p* > 0.05). There were no significant differences detected across any of the demographic data (age, gender, smoking status, family history) or clinical characteristics (PASI score and the DLQI) between the studied groups with *p* > 0.05, as shown in Table 1.

### 3.2. Baseline Laboratory Parameters Among the Studied Groups

The baseline laboratory assessments are represented in Table 2. No significant variations (*p* > 0.05) were found regarding the lipid profile, liver function tests, complete blood count, or inflammatory and oxidative stress markers between the three studied groups.

### 3.3. Follow-Up Laboratory Parameters Among the Studied Groups

#### 3.3.1. Liver Function Tests

The means ± SDs for alanine transaminase (ALT), aspartate transaminase (AST), and albumin levels throughout the study duration showed no significant changes among the studied groups (*p* > 0.05), as shown in Table 3.

Lipid Profile

Regarding the lipid profile parameters, the total cholesterol, triglycerides, low-density lipoprotein (LDL), and high-density lipoprotein (HDL) mean ± SD levels throughout the study duration showed no significant changes among the studied groups (*p* > 0.05), as shown in Table 3.

#### 3.3.2. Blood Glucose Test

Concerning the fasting blood glucose (FBG), the mean ± SD at baseline showed no significant differences among groups (control group: 86.14 ± 19.23; acetylcysteine group: 67.53 ± 14.91; and the acetylcysteine and Vitamin E group: 85.75 ± 19.51). In addition, throughout follow-ups 1, 2, and 3, no significant changes were observed among the studied groups (*p* > 0.05), as shown in Table 4.

#### 3.3.3. Complete Blood Count

The complete blood counts performed throughout the study included the platelet count, total leukocytic count (TLC), red blood cell (RBC) count, hemoglobin, hematocrit (HCT), mean corpuscular hemoglobin (MCH), mean corpuscular volume (MCV), and mean corpuscular hemoglobin concentration (MCHC), which showed no significant changes between the studied groups (*p* > 0.05), as shown in Table 4.

#### 3.3.4. Non-Specific Inflammatory Markers

Regarding the enrolled patients, there were no significant changes in the median values among the studied groups at baseline and during follow-ups 1 and 2 in the C-reactive protein (CRP) levels (*p* > 0.05); however, during follow-up 3, the acetylcysteine and Vitamin E group showed significantly lower median values when compared to the acetylcysteine group (*p* < 0.05). While the erythrocyte sedimentation rate (ESR) at baseline and at follow-ups 1 and 3 showed no significant changes in the medians of the studied groups (*p* > 0.05), as shown in Table 5, during follow-up 2, the acetylcysteine and Vitamin E group showed significantly lower median values when compared to the control group (*p* < 0.05).

### 3.4. Inflammatory Marker Interleukin-36 Gamma (IL-36γ) Levels Among the Studied Groups

The baseline mean ± SD of IL-36γ levels in the studied groups showed no statistical significance (*p* > 0.05). However, in follow-up 1, the mean ± SD in the control group was 113.8 ± 27.99, in the acetylcysteine group was 73.79 ± 25.99, which was significant from the control, and in the acetylcysteine and Vitamin E group was 68.66 ± 21.20, which was significant from the control (*p* < 0.05). Moreover, in follow-up 2, the mean ± SD in the control group was 101.90 ± 41.25, in the acetylcysteine group was 51.07 ± 10.57, which was significant from the control, and in the acetylcysteine and Vitamin E group was 52.45 ± 17.02, which was significant from the control (*p* < 0.05). In follow-up 3, the mean ± SD in the control group was 96.54 ± 49.27, in the acetylcysteine group was 38.06 ± 7.17, which was significant from the control, and in the acetylcysteine and Vitamin E group was 33.24 ± 9.49, which was significant from the control (*p* < 0.05).

When comparing the mean ± SD of IL-36γ in follow-up 3 versus baseline, both the acetylcysteine group and the acetylcysteine and Vitamin E group showed significant improvement in the IL-36 levels, as represented in Table 6 and Figure 2.

### 3.5. Oxidative Stress Marker Malondialdehyde (MDA) Levels Among the Studied Groups

The baseline mean ± SD of MDA levels in the studied groups showed no statistical significance, however, in follow-up 1, the mean ± SD in the control group was 0.89 ± 0.19, and in the acetylcysteine group was 0.58 ± 0.14, which was significantly lower than the control (*p* = 0.0003). The acetylcysteine and Vitamin E group was 0.68 ± 0.073, which was significant from the control (*p* = 0.039). Moreover, in follow-up 2, the mean ± SD in the control group was 0.80 ± 0.20, in the acetylcysteine group was 0.43 ± 0.072, which was significant from the control (*p* = 0.0001), and in the acetylcysteine and Vitamin E group was 0.55 ± 0.077, which was significant from the control (*p* = 0.0001). In follow-up 3, the mean ± SD in the control group was 0.76 ± 0.22, in the acetylcysteine group was 0.36 ± 0.052, which was significant from the control (*p* < 0.0001), and in the acetylcysteine and Vitamin E group was 0.37 ± 0.089, which was significant from the control (*p* < 0.0001).

When comparing the mean ± SD of MDA in follow-up 3 versus baseline, both the acetylcysteine group and the acetylcysteine and Vitamin E group showed significant improvement in the MDA levels (*p* < 0.0001), as represented in Table 7 and Figure 3.

### 3.6. Follow-Up Assessment of the Psoriasis Indices (PASI Score and the DLQI) Among the Studied Groups

As for the PASI score and the DLQI, the baseline scores in the studied groups showed no significant differences (*p* > 0.05), while at the end of the study duration, the treated groups (acetylcysteine group and acetylcysteine and Vitamin E group) showed significant improvement in the PASI score as compared to the control group (*p* < 0.05), where the acetylcysteine group showed a 42% average improvement and the acetylcysteine and Vitamin E group showed a 52% average improvement. Also, the PASI50 score was 63.33% and the PASI75 score was 23.33%. Additionally, the mean DLQIs in both the acetylcysteine (2.09 ± 2.84), and acetylcysteine and Vitamin E (2.33 ± 1.50) groups were significantly decreased compared to the control group (7.5 ± 6.07) and showed reductions by 3.6 and 4 points, respectively, as shown in Table 8 and Figure 4.

### 3.7. Correlations of IL-36γ and MDA with the PASI Score Among the Studied Groups

Table 9 shows a significant positive correlation between PASI and IL-36γ in all groups, both at baseline and after the third follow-up. Regarding the PASI and MDA, there were significant positive correlations in all groups, both at baseline and after the third follow-up.

### 3.8. Subject Presentation

Representations of psoriatic lesions are shown in Figure 5, Figure 6 and Figure 7, where in Figure 5 the control subject showed a marked worsening of the psoriatic lesion, and on the contrary, in Figure 6, the subject receiving acetylcysteine showed a marked decrease in the psoriatic lesion reaching the third follow-up, with almost complete healing of the lesion. Regarding the acetylcysteine and Vitamin E subject represented in Figure 7, the psoriatic lesion resolved completely by the third follow-up.

Assessment of Safety of NAC and Vitamin E

Usually, the side effects of oral NAC are minimal. Gastrointestinal symptoms, including nausea and vomiting, are the most experienced adverse effects after oral administration of NAC. In most cases, they resolved within 3–4 days. The strong sulfur-based odor of NAC, which makes it smell like rotten eggs, contributes to the nausea and vomiting symptoms that occur after oral administration. NAC is usually diluted in caffeine-free diet beverages to mask its taste and odor and to make it more palatable [29]. Additionally, Vitamin E is generally considered safe when used as a supplement in doses not greater than 1000 mg/day [39,40]. Therefore, none of the patients discontinued the therapy.

## 4. Discussion

Currently, the global community believes that preventive medicine should be integrated into treatment and follow-up plans, particularly for chronic diseases. Chronic diseases often significantly affect a patient’s lifestyle, career, and social life. By incorporating preventive strategies, the progression of the disease can be slowed, improving the patient’s quality of life and reducing the burden on their daily activities. Preventive medicine aims to stabilize chronic conditions over the long term, reducing complications and hospitalizations. One of the key benefits of preventive medicine is slowing the progression of chronic diseases. This can prevent the development of additional comorbidities (other diseases or conditions), which are common in patients with chronic illnesses [41].

As psoriasis is a chronic disease, like hypertension, cardiac diseases, or diabetes, patients need to be well-educated about the disease, its nature, and its mechanism. This will help them accept it and develop effective coping strategies [42]. Psoriasis is fundamentally an inflammatory disease associated with high levels of free radicals in the body. If the inflammatory mediators are successfully suppressed and oxidative stress levels are reduced, psoriasis can be effectively controlled. This would result in fewer flare-ups, which, if managed early, can be easily controlled [43]. Typically, treatment is customized to address the specific symptoms a patient experiences, such as itching, redness, pain, and scaling. Conventional treatments usually include topical applications, such as steroids, other anti-inflammatory agents like calcipotriene, tazarotene, or tacrolimus, and topical salicylic acid to reduce scaling. The aim of this study was to explore an alternative treatment option that could be beneficial for mild to moderate psoriasis vulgaris cases.

In the current study, N-acetylcysteine (NAC), a powerful antioxidant and anti-inflammatory agent, was chosen to be added to the treatment plan [44]. Vitamin E was also included as a skin antioxidant [31]. The efficacy of this add-on therapy was evaluated both clinically and through laboratory measurements. Additionally, its impact on patients’ daily routines and quality of life was reassessed.

As shown in our results, significant improvement was observed in the groups receiving acetylcysteine—either alone or in combination with Vitamin E—in addition to their conventional treatment, compared to the control group, which received only standard conventional treatment. This improvement was evident in clinical manifestations as measured by the PASI (Psoriasis Area and Severity Index), in quality of daily life activities as assessed by the DLQI (Dermatology Life Quality Index), and in the levels of inflammation and oxidative stress in the body.

As for the PASI and DLQI scores, the baseline scores in the studied groups showed no significant differences (*p* > 0.05). However, at the end of three months, the treated groups (the acetylcysteine group and the acetylcysteine and Vitamin E group) showed significant improvement in both scores compared to the control group (*p* < 0.05). Furthermore, the acetylcysteine and Vitamin E group showed the highest PASI average improvement of 52%, while the acetylcysteine group alone showed a PASI average improvement of 41%. In comparison, the control group showed no improvement and an average worsening in PASI by 6%.

This finding is consistent with the results of a pilot study, which assessed improvement of psoriasis patients using a non-denatured whey protein that enhances glutathione, which demonstrated that PASI scores improved after consuming a source of antioxidants. This is likely due to the enhancement of glutathione levels in the body, leading to a potent free radical scavenging effect [45]. Moreover, psoriasis vulgaris is an autoimmune disease characterized by inflammatory attacks and high levels of free radicals. The addition of a powerful antioxidant and anti-inflammatory agent, such as acetylcysteine, can significantly reduce uncontrolled skin inflammation [46].

The serum MDA concentration in the control group was reported to be 1.08 ± 0.27 nmol/mL at baseline and 0.86 ± 0.19 nmol/mL after 12 weeks, indicating no change over time (*p*  =  0.154). On the other hand, the serum MDA concentration in the acetylcysteine group was reported to be 1.05 ± 0.28 nmol/mL at baseline and 0.36 ± 0.052 nmol/mL after 12 weeks, indicating a strong decline of oxidative stress upon NAC treatment (*p*  <  0.001)

Fundamentally, NAC is believed to be a potent antioxidant due to its role in glutathione replenishment. When it enters the body, deacetylation occurs, releasing the amino acid cysteine, which is a precursor of glutathione. The enzyme γ-glutamyl cysteine ligase (γ-GCL) conjugates L-cysteine with L-glutamate to form γ-glutamyl cysteine (γ-GC). L-glycine is then incorporated to form the final glutathione (GSH) tripeptide [47]. Since glutathione plays a critical role in the catalytic reduction of disulfides and peroxides, its depletion leads to elevated levels of malondialdehyde, the end product of lipid peroxidation [48].

Additionally, one of the mechanisms behind NAC’s cytoprotective effect involves the production of sulfane sulfur species through desulfuration. This process leads to the activation of enzymes such as 3-mercaptopyruvate sulfur transferase (MST) and/or sulfide quinone oxidoreductase (SQR). Therefore, it is evident that NAC has the potential to prevent or at least mitigate the damage to cells that results from the unforeseen oxidation process. It clearly helps reload the body with the most potent detoxifying amino acid, glutathione, which is not only good for the liver and kidney, but also in various conditions like asthma, COPD, ulcerative colitis, and infertility, and also helps in preventing some cardiovascular events happening due to oxidation of the heart cells, especially with diabetes [49,50,51].

Furthermore, NAC demonstrates potent antioxidant and anti-inflammatory properties, which lead to a substantial improvement in acetic acid-induced colitis in rats. It inhibits nuclear factor kappa B (NF-κB), a pro-inflammatory transcription factor activated by high levels of reactive oxygen species. Additionally, NAC mitigates the chemotactic activity of neutrophils by reducing levels of lactoferrin, anti-chymotrypsin, and eosinophil cationic protein (ECP). It also significantly lowers inflammatory biomarkers such as IL-6, TNF-α, IL-10, and IFN-γ [44].

This is in accordance with other studies that investigated the efficacy of oral curcumin, a natural anti-inflammatory agent. Oral curcumin was used in the treatment of moderate to severe psoriasis vulgaris in a prospective clinical trial. The addition of oral curcumin led to an impressive improvement in PASI score, as represented by the PASI75 score at the end of the study [52].

Vitamin E is considered the superior physiological barrier antioxidant in human skin cells. It protects skin cells from oxidative damage caused by internal conditions or external factors, such as direct sunlight. Vitamin E reduces acute skin reactions to sunlight and minimizes adverse skin events related to autoimmune processes [53]. As a potent antioxidant, it inhibits lipid peroxidation and reduces the conversion of arachidonic acid to PGE2, a pro-inflammatory mediator. This helps slow down the ongoing inflammation associated with autoimmune diseases [54].

In autoimmune diseases, the level of IgE typically increases due to overactivity of the immune system. According to previous studies, serum IgE levels are significantly higher in patients with psoriasis as compared to controls. It has been suggested that Vitamin E supplementation can reduce IgE levels associated with autoimmune diseases, thereby alleviating reactions caused by an overactive immune response to allergens [55,56].

Additionally, a randomized controlled study evaluated the impact of Vitamin E and conjugated linoleic acid, either separately or in combination, on inflammatory and immune markers among subjects with active rheumatoid arthritis. The results indicated that vitamin E supplementation significantly suppressed pro-inflammatory cytokines and modulated cyclooxygenase-2 activity [57]. However, studies have recently highlighted that vitamin E absorption is rather complex and requires particular intestinal transport proteins, accounting for a variable absorption of vitamin E from 10 to 79% and affecting its bioavailability [58,59].

In the current study, the serum IL-36γ concentration in the control group was reported to be 122.7 ± 34.40 pg/mL at baseline and 96.54 ± 49.27 pg/mL after 12 weeks, indicating no change over time (*p*  =  0.154). However, the serum IL-36γ concentration in the acetylcysteine and Vitamin E group was reported to be 95.89 ± 31.17 pg/mL at baseline and 33.24 ± 9.49 pg/mL after 12 weeks, indicating a significant anti-inflammatory effect of NAC with Vitamin E (*p*  <  0.001).

The IL-36 family of cytokines consists of three agonists (α, β, and γ) and one antagonist (Ra). The two main sources of IL-36 cytokines in the skin are keratinocytes and immune system cells. It has been shown that several cytokines were associated with psoriasis, including TNFα and IL-22, which both, either alone or in combination, promote IL-36 production by primary human keratinocytes and organotypic skin models. The cytokines themselves also activate or boost the production of the IL-36 gene in cells; IL-36R activation is the most potent inducer of IL-36γ [60].

The IL-36 gene is expressed by human monocytes, macrophages, CD4 + T cells, and dendritic cells derived from mouse bone marrow in addition to keratinocytes. The protein expression of the IL-36 agonist was verified by immunohistochemical labeling and increased mRNA expression of IL-36 cytokines, particularly IL-36Ra, in psoriatic skin lesions in various cells such as keratinocytes, endothelial cells, dermal fibroblasts, dendritic cells, and CD68 + macrophages.

In mice with overexpressed IL-36α in keratinocytes but an otherwise normal phenotype, 12-O-tetradecanoylphorbol-13-acetate (TPA) treatment results in a considerable increase in skin inflammation that morphologically mimics psoriasis, as well as a significant rise in IL-17A, IL-22, and IL-23. Using an IL-36R blocking antibody reduced inflammation in this mouse model of psoriasis, demonstrating its dependence on IL-36R signaling. Additionally, it reacted well to treatments that disrupt three pathways that are effectively addressed in psoriasis: TNFα, IL-12/23p40, and IL-23p19. These results demonstrate that IL-36 is a crucial mediator of psoriatic inflammation in this scenario [61].

Additionally, a previous study [62] investigated the impact of oral omega-3 fatty acid supplementation on inflammatory processes in different inflammatory diseases like rheumatoid arthritis, Crohn’s disease, lupus, and psoriasis. The study observed a significant reduction in the inflammatory cytokines TNF-α, IL-1β, and IL-6. Furthermore, it detected an inhibition in the activation of nuclear factor κ-B (NF-κB), a transcription factor for B cells.

The current study also observed that the inflammatory CRP mean levels in the control, acetylcysteine, and acetylcysteine and Vitamin E groups were 0.255, 0.400, and 0.211 g/dL, respectively, with no significance at baseline. Furthermore, no significant difference in mean levels was observed among the three groups after F1 (control: 0.425, acetylcysteine: 0.418, and acetylcysteine and Vitamin E: 0.362 g/dL) and F2 (control: 0.376, acetylcysteine: 0.341, and acetylcysteine and Vitamin E: 0.329 g/dL). Nevertheless, after F3, the CRP mean level in the acetylcysteine and Vitamin E group (0.157 g/dL) was significantly reduced in comparison with the acetylcysteine group (0.331 g/dL).

CRP is an acute phase reactant, widely recognized as a non-specific systemic inflammatory biomarker, and was proven to be elevated in cases of active psoriasis [63]. Although its levels were correlated with the severity of psoriasis, its levels are not uniquely specific to psoriasis and could be influenced by any other cause of inflammation [64]. CRP levels possess the capability to be a simple and inexpensive marker for determining the inflammatory status in psoriasis patients and demonstrating a higher inflammatory load as compared to healthy individuals [64,65]. In addition, a cross-sectional study correlated elevated CRP levels in psoriasis patients with depression [66]. The slight decrease in CRP could be attributed to several factors, such as the non-specificity of CRP, where other comorbidities may influence its levels [67], a genetic predisposition that may affect CRP levels, variability in the immune system’s release of CRP upon activation, and lifestyle differences [68,69]. Another study utilized NAC in the treatment of a chronic respiratory condition, and a significant reduction in CRP was observed only with a high dose of 1200 mg/day [70].

This study explores a novel add-on therapy that could be beneficial for mild to moderate psoriasis vulgaris by targeting both inflammatory and oxidative stress processes within the body. The therapy involves readily available and cost-effective products that may help reduce the duration of flare-ups and prevent worsening of the condition. Additionally, it has the potential to stabilize the condition for longer periods, resulting in fewer flare episodes. However, our study has a short duration and a small sample size, and while the current study demonstrates promising results with N-acetylcysteine and Vitamin E in managing psoriasis vulgaris, it is important to note that the long-term effects and potential recurrence of symptoms after discontinuation of therapy were not evaluated. Unlike topical steroids, which are known to cause rebound flares upon withdrawal due to immune suppression, the antioxidant and anti-inflammatory mechanisms of NAC and Vitamin E may carry a lower risk of such adverse effects. Nevertheless, further studies with extended follow-up periods are warranted to investigate the durability of treatment outcomes and the risk of disease recurrence after therapy cessation. Additionally, we also recommend additional multicenter, long-term duration future studies to evaluate the safety and efficacy of long-term therapy. Moreover, we recommend a study of the effects of a topical nanoformulation of vitamin E in the management of psoriatic lesions.

## 5. Conclusions

Our study proved that the addition of N-acetylcysteine to the treatment plan of psoriasis patients resulted in a significant reduction in the inflammatory load, represented by IL-36γ, and in the oxidative stress level, represented by MDA. Moreover, a significant improvement in the clinical manifestation has been shown, represented by a PASI score change, along with improvement in the quality of life as represented by DLQI. However, the addition of Vitamin E did not show a significant difference when added to N-acetylcysteine, but the patients well-tolerated it due to its excellent effect on the skin, as it helps greatly in controlling the disease’s clinical manifestation.

## Figures and Tables

**Figure 1 biomedicines-13-01275-f001:**
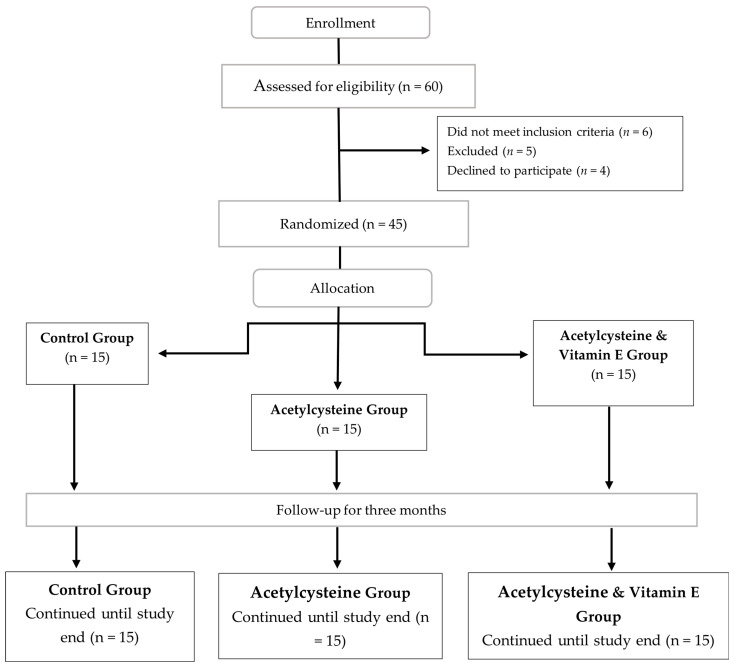
CONSORT flow diagram for the enrolled participants with mild to moderate psoriasis vulgaris.

**Figure 2 biomedicines-13-01275-f002:**
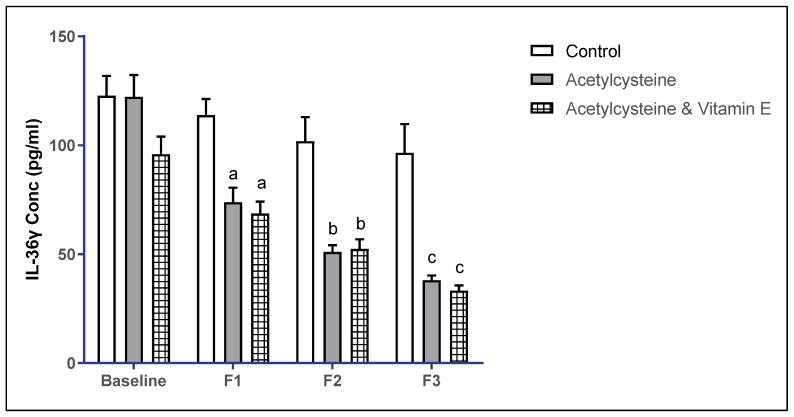
Multiple comparisons of IL-36γ throughout the study duration among the studied groups. Significance threshold *p* < 0.05; a: significant from control F1, b: significant from control F2, c: significant from control F3.

**Figure 3 biomedicines-13-01275-f003:**
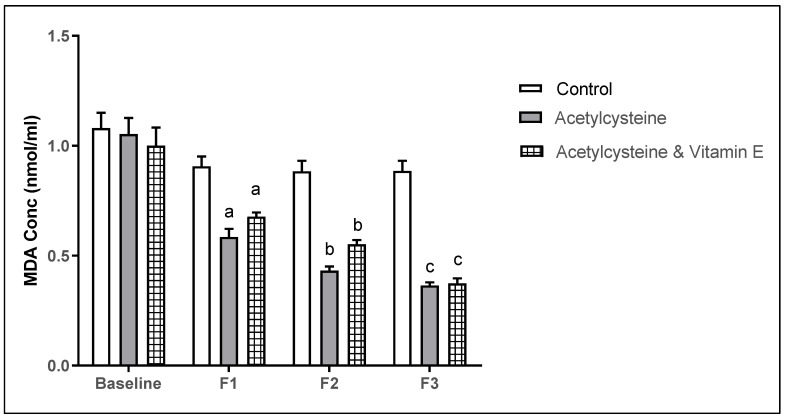
Multiple comparisons of MDA throughout the study duration among the studied groups. (Significance threshold *p* < 0.05. a: significant from control F1, b: significant from control F2, c: significant from control F3).

**Figure 4 biomedicines-13-01275-f004:**
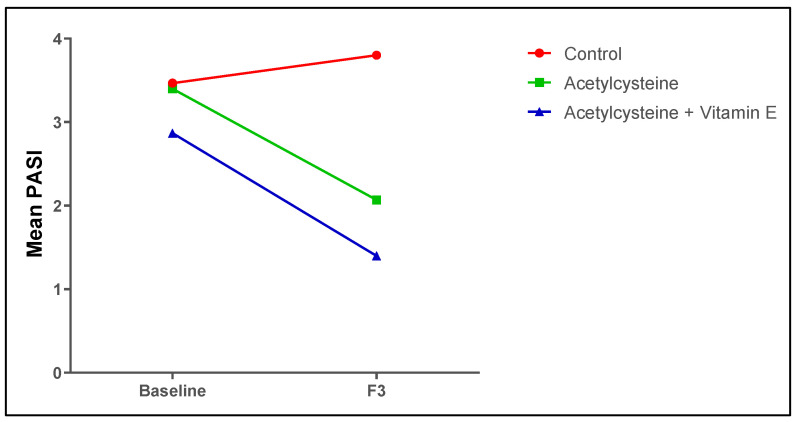
Mean PASI scores among the studied groups at baseline and at the end of the study.

**Figure 5 biomedicines-13-01275-f005:**
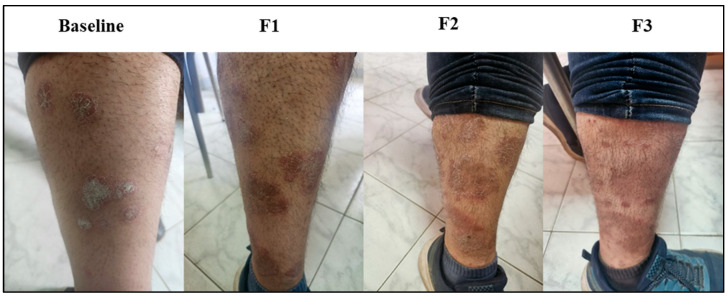
Psoriatic lesion representation of the control group at baseline, follow-up 1 (F1), follow-up 2 (F2), and follow-up 3 (F3).

**Figure 6 biomedicines-13-01275-f006:**
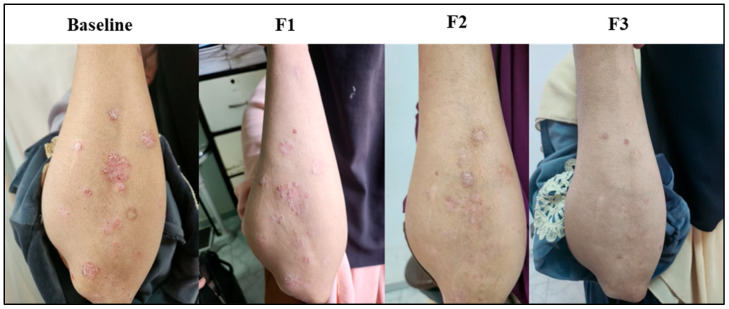
Psoriatic lesion representation of the acetylcysteine group at baseline, follow-up 1 (F1), follow-up 2 (F2), and follow-up 3 (F3).

**Figure 7 biomedicines-13-01275-f007:**
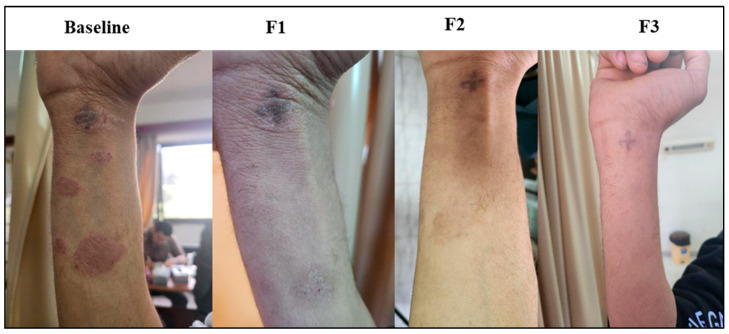
Psoriatic lesion representation of the acetylcysteine and Vitamin E group at baseline, follow-up 1 (F1), follow-up 2 (F2), and follow-up 3 (F3).

**Table 1 biomedicines-13-01275-t001:** Patients’ demographics and baseline characteristics among the studied groups.

Variables		Control (n = 15)	Acetylcysteine (n = 15)	Acetylcysteine and Vitamin E (n = 15)	P1-Value	P2-Value		P3-Value
Demographics								
**Age (years)**	Mean ± SD	43.10 ± 8.3	40.00 ± 12.65	41.93 ± 9.30	0.99 ^#^	0.94 ^#^		0.94 ^#^
**Sex (n)**	Male	6 (40%)	8 (43.33%)	13 (86.67%)	0.027 ^$^*			
	Female	9 (60%)	7 (46.67%)	2 (13.33%)				
**Smoking status (n)**	Smoker	3 (6.67%)	2 (4.44%)	3 (6.67%)	0.86 ^$^			
	Non-smoker	12 (26.67%)	13 (28.89%)	12 (26.67%)				
**Family History** **(n)**	Yes	3	2	0	0.21 ^$^			
	No	12	13	15				
**PASI score**	Mean ± SD	3.47 ± 0.64	3.40 ± 0.63	2.87 ± 0.92	>0.999 ^#^	0.697 ^#^	0.791 ^#^	
**DLQI**	Mean ± SD	4.25 ± 2.87	5.64 ± 3.59	5.91 ± 2.28	0.959 ^#^	0.913 ^#^	>0.999 ^#^	

PASI: Psoriasis Area and Severity Index; DLQI: Dermatology Life Quality Index. Statistical tests: #: Tukey HSD; $: chi-square. * Statistically significant (*p*-value < 0.05). P1-value: acetylcysteine vs. control; P2-value: acetylcysteine and Vitamin E vs. control; P3-value: acetylcysteine and Vitamin E vs. acetylcysteine.

**Table 2 biomedicines-13-01275-t002:** Baseline values of the laboratory parameters for the studied groups.

Variables	Control (n = 15)	Acetylcysteine (n = 15)	Acetylcysteine and Vitamin E (n = 15)	P1-Value	P2-Value	P3-Value
**Routine Lab Analysis**						
**AST** (U/L) Mean ± SD	26.43 ± 8.31	28.47 ± 5.44	22.60 ± 6.25	0.0646 ^$$^		
**ALT** (U/L) Mean ± SD	23.20 ± 5.69	26.87 ± 7.98	21.93 ± 5.74	0.117 ^$$^		
**Albumin** (g/dL) Mean ± SD	4.14 ± 0.44	4.19 ± 0.45	4.16 ± 0.47	0.966 ^$$^		
**Total cholesterol** (mg/dL) Mean ± SD	149.00 ± 30.80	154.00 ± 32.00	129.00 ± 28.20	0.0716 ^$$^		
**TG** (mg/dL) Mean ± SD	99.80 ± 25.4	114.90 ± 18.72	94.14 ± 24.59	0.0610 ^$$^		
**LDL** (mg/dL) Mean ± SD	86.31 ± 30.51	93.04 ± 31.59	78.64 ± 31.55	0.510 ^$$^		
**HDL** (mg/dL) Mean ± SD	40.80 ± 3.56	40.30 ± 5.87	41.90 ± 3.71	0.689 ^$$^		
**FBG** (mg/dL) Mean ± SD	86.14 ± 19.23	67.53 ± 14.91	85.75 ± 19.51	0.106 ^$$^	>0.99 ^$$^	0.104 ^$$^
**Complete Blood Count**						
**Platelets** (×10^3^/µL) Median (range)	300.50 (35.4–414)	272.40 (227–385.2)	345.00 (194–433)	0.410 ^##^		
**TLC count** (×10^3^/µL) Median (range)	8.10 (5.1–10.1)	8.50 (4.9–9.7)	7.80 (3.7–10.3)	0.822 ^##^		
**RBC count** (×10^3^/µL) Median (range)	6.88 (4–8.18)	5.50 (3.5–7.62)	7.24 (4.6–8.94)	0.0614 ^##^		
**Hemoglobin** (g/dL) Mean ± SD	13.16 ± 1.39	13.47 ± 1.58	14.08 ± 1.70	0.281 ^$$^		
**HCT** (%) Median (range)	39.80 (31.6–46.2)	39.30 (31.3–54.4)	41.70 (30.1–4.9)	0.158 ^##^		
**MCV** (fl) Median (range)	49.25 (42.0–58.0)	50.00 (42.7–58.0)	52.70 (43.0–61)	0.064 ^##^		
**MCH** (pg) Median (range)	15.75 (11.5–21.5)	14.10 (12.0–25.5)	18.70 (12.5–28.1)	0.824 ^##^	0.072 ^##^	0.02 ^##^*
**MCHC** (g/dL) Median (range)	34.05 (28.0–37.5)	36.10 (30.0–38.6)	35.20 (32.0–36.0)	0.0699 ^##^		
**Inflammatory Markers**						
**CRP** (g/dL) Median (range)	0.255 (0.0410–0.428)	0.400 (0.062–0.520)	0.211 (0.060–0.50)	0.0816 ^##^		
**ESR (mm/h)** **Median (range)**	12 (3–35)	5 (3–30)	15 (5–50)	0.116 ^##^		
**IL-36** (pg/mL) Mean ± SD	122.70 ± 34.40	122.20 ± 39.06	95.89 ± 31.17	>0.999 ^^^	0.351 ^^^	0.381 ^^^
**Oxidative Stress Marker**						
**MDA** (nmol/mL) Mean ± SD	1.08 ± 0.27	1.05 ± 0.28	1.00 ± 0.32	>0.999 ^^^	0.989 ^^^	0.999 ^^^

ALT: alanine transaminase; AST: aspartate transaminase; TG: triglycerides; LDL: low-density lipoprotein; HDL: high-density lipoprotein; CRP: C-reactive protein; ESR: erythrocyte sedimentation rate; FBG: fasting blood glucose; TLC: total leukocyte count; RBCs: red blood cells; HCT: hematocrit; MCV: mean corpuscular volume; MCH: mean corpuscular hemoglobin; MCHC: mean corpuscular hemoglobin concentration. Statistical tests: ## Kruskal–Wallis Test; $$ post-hoc test, one-way ANOVA. ^: two-way ANOVA and Tukey post-hoc test. * Statistically significant (*p*-value < 0.05). P1-value: acetylcysteine vs. control; P2-value: acetylcysteine and Vitamin E vs. control; P3-value: acetylcysteine and Vitamin E vs. acetylcysteine.

**Table 3 biomedicines-13-01275-t003:** Follow-up laboratory assessment of liver function and lipid profile among the studied groups.

Variables		Control (n = 15)	Acetylcysteine (n = 15)	Acetylcysteine and Vitamin E (n = 15)	P1-Value	P2-Value	P3-Value
**Liver Function tests**							
**AST** (U/L) Mean ± SD	Baseline	26.43 ± 8.31	28.47 ± 5.44	22.60 ± 6.25	0.0646 ^$$^		
	F1	25.07 ± 6.22	28.20 ± 6.67	25.33 ± 5.14	0.305 ^$$^		
	F2	26.33 ± 4.52	25.33 ± 4.68	25.87 ± 4.63	0.869 ^$$^		
	F3	28.33 ± 5.20	27.45 ± 4.55	25.50 ± 6.11	0.465 ^$$^		
**ALT** (U/L) Mean ± SD	Baseline	23.20 ± 5.69	26.87 ± 7.98	21.93 ± 5.74	0.117 ^$$^		
	F1	27.86 ± 6.18	28.07 ± 6.82	28.87 ± 4.98	0.892 ^$$^		
	F2	27.17 ± 5.46	30.08 ± 6.50	25.07 ± 6.34	0.122 ^$$^		
	F3	28.78 ± 6.82	26.64 ± 5.84	25.42 ± 5.16	0.440 ^$$^		
**Albumin** (g/dL) Mean ± SD	Baseline	4.14 ± 0.44	4.19 ± 0.45	4.16 ± 0.47	0.966 ^$$^		
	F1	4.21 ± 0.51	4.09 ± 0.53	4.18 ± 0.46	0.796 ^$$^		
	F2	4.27 ± 0.53	4.28 ± 0.52	4.13 ± 0.46	0.670 ^$$^		
	F3	4.13 ± 0.42	4.01 ± 0.56	4.21 ± 0.34	0.566 ^$$^		
**Lipid Profile**							
**Total cholesterol** (mg/dL) Mean ±SD	Baseline	149.00 ± 30.80	154.00 ± 32.00	129.00 ± 28.20	0.0716 ^$$^		
	F1	142.80 ± 27.92	125.50 ± 23.19	131.90 ± 30.57	0.241 ^$$^		
	F2	141.00 ± 29.90	162.70 ± 25.45	145.00 ± 24.29	0.113 ^$$^		
	F3	140.00 ± 38.45	155.00 ± 13.04	157.50 ± 16.75	0.233 ^$$^		
**TG** (mg/dL) **Mean ± SD**	Baseline	99.80 ± 25.4	114.90 ± 18.72	94.14 ± 24.59	0.0610 ^$$^		
	F1	130.40 ± 24.36	129.50 ± 20.92	138.70 ± 26.16	0.367 ^$$^		
	F2	128.20 ± 33.64	123.90 ± 24.31	132.50 ± 28.28	0.748 ^$$^		
	F3	123.80 ± 21.00	125.40 ± 36.12	113.20 ± 26.22	0.555 ^$$^		
**LDL** (mg/dL) **Mean ± SD**	Baseline	86.31 ± 30.51	93.04 ± 31.59	78.64 ± 31.55	0.510 ^$$^		
	F1	76.64 ± 29.18	61.97 ± 20.78	63.45 ± 31.65	0.312 ^$$^		
	F2	75.57 ± 29.69	97.30 ± 23.29	77.31 ± 26.35	0.0917 ^$$^		
	F3	76.24 ± 36.14	90.67 ± 13.90	93.54 ± 14.32	0.205 ^$$^		

F1: follow-up 1, F2: follow-up 2, F3: follow-up 3. ALT: alanine transaminase; AST: aspartate transaminase; TG: triglycerides; LDL: low-density lipoprotein; HDL: high-density lipoprotein. Statistical tests: $$ post-hoc test, one-way ANOVA. P1-value: acetylcysteine vs. control; P2-value: acetylcysteine and Vitamin E vs. control; P3-value: acetylcysteine and Vitamin E vs. acetylcysteine.

**Table 4 biomedicines-13-01275-t004:** Follow-up laboratory assessment of fasting blood glucose and complete blood count among the studied groups.

Variables		Control (n = 15)	Acetylcysteine (n = 15)	Acetylcysteine and Vitamin E (n = 15)	P1-Value	P2-Value	P3-Value
**Blood Glucose test**							
**FBG** (mg/dL) **Mean ± SD**	Baseline	86.14 ± 19.23	67.53 ± 14.91	85.75 ± 19.51	0.106 ^$$^	>0.99 ^$$^	0.104 ^$$^
	F1	87.50 ± 22.03	74.47 ± 14.26	70.09 ± 7.006	0.497 ^$$^	0.189 ^$$^	0.99 ^$$^
	F2	76.75 ± 14.51	71.08 ± 16.37	67.20 ± 5.92	0.999 ^$$^	0.906 ^$$^	>0.99 ^$$^
	F3	77.00 ± 10.64	77.91 ± 11.14	78.58 ± 17.87	>0.99 ^$$^	>0.99 ^$$^	>0.99 ^$$^
Complete Blood Count							
**Platelets** (×10^3^/µL) Median (range)	Baseline	300.50 (35.4–414)	272.41 (227–385.2)	345.00 (194–433)	0.410 ^##^		
	F1	291.50 (233.5–410)	307.00 (246–410)	273.60 (205–417)	0.579 ^##^		
	F2	332.00 (205–417)	312.00 (234–410)	328.00 (251–417)	0.711 ^##^		
	F3	294.00 (35.2–410)	297.00 (34.6–433)	355.50 (36.5–414)	0.0816 ^##^		
**TLC count** (×10^3^/µL) Median (range)	Baseline	8.10 (5.1–10.1)	8.50 (4.9–9.7)	7.80 (3.7–10.3)	0.822 ^##^		
	F1	6.50 (4.1–9.4)	6.20 (4.5–9.7)	5.90 (4.1–8.3)	0.455 ^##^		
	F2	6.35 (5.6–9.7)	5.66 (4.15–10.1)	6.40 (4–10.3)	0.656 ^##^		
	F3	6.30 (5.1–9.7)	8.10 (4.15–9.7)	6.00 (4.2–9.4)	0.326 ^##^		
**RBC count** (×10^3^/µL) Median (range)	Baseline	6.88 (4–8.18)	5.5 (3.5–7.62)	7.24 (4.6–8.94)	0.0614 ^##^		
	F1	4.40 (3.5–7.3)	5.10 (3.9–9.6)	4.30 (3.6–8.7)	0.686 ^##^		
	F2	6.35 (5.6–8.9)	5.69 (3.7–8.4)	5.50 (4.15–8.4)	0.050 ^##^		
	F3	4.50 (3.8–7.86)	5.90 (5.1–8.8)	5.90 (4.5–7.2)	0.255 ^##^		
**Hemoglobin** (g/dL) **Mean ± SD**	Baseline	13.16 ± 1.39	13.47 ± 1.58	14.08 ± 1.70	0.281 ^$$^		
	F1	13.16 ± 1.04	12.91 ± 1.50	13.55 ± 1.14	0.385 ^$$^		
	F2	13.47 ± 1.36	13.33 ± 1.36	13.25 ±0.91	0.900 ^$$^		
	F3	13.40 ± 1.79	13.62 ± 1.57	13.07 ± 0.95	0.655 ^$$^		
**HCT** (%) Median (range)	Baseline	39.80 (31.6–46.2)	39.30 (31.3–54.4)	41.70 (30.1–4.9)	0.158 ^##^		
	F1	35.85 (28.4–43.3)	34.30 (30.4–50.0)	36.20 (30.3–49.5)	0.970 ^##^		
	F2	35.30 (31.6–54.4)	38.40 (33.8–48.0)	40.00 (36.0–48.0)	0.051 ^##^		
	F3	36.40 (31.3–54.4)	37.20 (33.6–47.9)	41.10 (31.8–46.2)	0.471 ^##^		
**MCV** (fl) Median (range)	Baseline	49.25 (42.0–58.0)	50.00 (42.7–58.0)	52.70 (43.0–61)	0.064 ^##^		
	F1	50.85 (43.2–56.2)	50.00 (43.2–55.0)	49.50 (43.7–57.0)	0.839 ^##^		
	F2	51.1 (43.2–55.0)	52.0 (49.0–56.0)	52.0 (41.0–59.0)	0.418 ^##^		
	F3	52.5 (43.7–58.0)	51.2 (44.0–58.0)	49.9 (41.8–56.3)	0.833 ^##^		
**MCH** (pg) Median (range)	Baseline	15.75 (11.5–21.5)	14.10 (12.0–25.5)	18.70 (12.5–28.1)	0.824 ^##^	0.072 ^##^	0.02 ^##^*
	F1	14.45 (11.6–19.3)	18.40 (12.1–21.3)	16.50 (12.0–19.3)	0.029 ^##^*	>0.99 ^##^	0.196 ^##^
	F2	17.90 (13.3–21.7)	14.50 (11.7–18.3)	14.90 (13.2–18.3)	0.018 ^##^*	0.017 ^##^*	>0.99 ^##^
	F3	16.50 (12.6–21.7)	15.60 (11.0–18.7)	17.20 (14.4–21.4)	0.19 ^##^		
**MCHC** (g/dL) Median (range)	Baseline	34.05 (28.0–37.5)	36.10 (30.0–38.6)	35.20 (32.0–36.0)	0.0699 ^##^		
	F1	30.00 (25.0–37.0)	32.00 (26.0–38.5)	30.00 (25.0–36.0)	0.172 ^##^		
	F2	31.50 (26.0–37.0)	32.40 (29.0–39.0)	33.00 (30.0–39.0)	0.411 ^##^		
	F3	33.80 (26.0–36.2)	33.50 (30.0–39.0)	35.30 (30.0–39.2)	0.337 ^##^		

F1: follow-up 1, F2: follow-up 2, F3: follow-up 3. FBG: fasting blood glucose; TLC: total leukocyte count; RBCs: red blood cells; HCT: hematocrit; MCV: mean corpuscular volume; MCH: mean corpuscular hemoglobin; MCHC: mean corpuscular hemoglobin concentration. Statistical tests: ## Kruskal–Wallis Test; $$ post-hoc test, one-way ANOVA. * Statistically significant (*p*-value < 0.05). P1-value: acetylcysteine vs. control; P2-value: acetylcysteine and Vitamin E vs. control; P3-value: acetylcysteine and Vitamin E vs. acetylcysteine.

**Table 5 biomedicines-13-01275-t005:** Follow-up laboratory assessment of non-specific inflammatory markers among the studied groups.

Variables		Control (n = 15)	Acetylcysteine (n = 15)	Acetylcysteine and Vitamin E (n = 15)	P1-Value	P2-Value	P3-Value
Non-specific Inflammatory Markers							
**CRP** (g/dL) Median (range)	Baseline	0.255 (0.0410–0.428)	0.400 (0.062–0.520)	0.211 (0.060–0.50)	0.0816 ^##^		
	F1	0.425 (0.344–0.538)	0.418 (0.242–0.538)	0.362 (0.161–0.638)	0.263 ^##^		
	F2	0.376 (0.066–0.662)	0.341 (0.271–0.638)	0.329 (0.131–0.583)	0.844 ^##^		
	F3	0.390 (0.037–0528)	0.331 (0.162–0.53)	0.157 (0.057–0.400)	>0.99 ^##^	0.123 ^##^	0.0145 *^##^
	**P4-value**	0.8351 ^	>0.999 ^	0.708 ^			
**ESR (mm/h)** Median (range)	Baseline	12 (3–35)	5 (3–30)	15 (5–50)	0.116 ^##^		
	F1	17.5 (2–35)	10 (3–30)	10 (5–15)	0.303 ^##^		
	F2	16 (6–35)	10 (3–20)	5 (3–45)	0.05 ^##^	0.010 *^##^	>0.99 ^##^
	F3	10 (3–40)	10 (3–20)	5 (3–33)	0.209 ^##^		
	**P4-value**	0.993 ^	>0.999 ^	0.159 ^			

F1: follow-up 1, F2: follow-up 2, F3: follow-up 3. CRP: C-reactive protein; ESR: erythrocyte sedimentation rate. Statistical tests: ## Kruskal–Wallis Test; ^: two-way ANOVA and Tukey post-hoc test. * Statistically significant (*p*-value < 0.05). P1-value: acetylcysteine vs. control; P2-value: acetylcysteine and Vitamin E vs. control; P3-value: acetylcysteine and Vitamin E vs. acetylcysteine.

**Table 6 biomedicines-13-01275-t006:** Follow-up laboratory assessment of IL-36γ levels among the studied groups.

Variables		Control (n = 15)	Acetylcysteine (n = 15)	Acetylcysteine and Vitamin E (n = 15)	P1-Value	P2-Value	P3-Value
Inflammatory Marker IL-36γ							
**IL-36γ** (pg/mL) **Mean ± SD**	Baseline	122.70 ± 34.40	122.20 ± 39.06	95.89 ± 31.17	>0.999 ^^^	0.351	0.381
	F1	113.80 ± 27.99	73.79 ± 25.99	68.66 ± 21.20	0.014 *^^^	0.0025 *^^^	>0.999 ^^^
	F2	101.90 ± 41.25	51.07 ± 10.57	52.45 ± 17.02	0.0004 *^^^	0.0005 *^^^	>0.999 ^^^
	F3	96.54 ± 49.27	38.06 ± 7.17	33.24 ± 9.49	0.0001 *^^^	<0.0001 *^^^	>0.999 ^^^
	**P4-value**	0.154	<0.0001 *^^^	<0.0001 *^^^			

IL-36: interleukin-36. Statistical tests: ^: two-way ANOVA and Tukey post-hoc test. * Statistically significant (*p*-value < 0.05). P1-value: acetylcysteine vs. control; P2-value: acetylcysteine and Vitamin E vs. control; P3-value: acetylcysteine and Vitamin E vs. acetylcysteine; P4-value: F3 vs. baseline.

**Table 7 biomedicines-13-01275-t007:** Follow-up laboratory assessment of MDA levels among the studied groups.

Variables		Control (n = 15)	Acetylcysteine (n = 15)	Acetylcysteine and Vitamin E (n = 15)	P1-Value	P2-Value	P3-Value
Oxidative Stress Marker (MDA)							
**MDA** (nmol/mL) **Mean ± SD**	Baseline	1.08 ± 0.27	1.05 ± 0.28	1.00 ± 0.32	>0.999 ^^^	0.989 ^^^	0.999 ^^^
	F1	0.89 ± 0.19	0.58 ± 0.14	0.68 ± 0.073	0.0003 *^^^	0.039 *^^^	0.966 ^^^
	F2	0.88 ± 0.18	0.43 ± 0.072	0.55 ± 0.077	0.0001 *^^^	0.0001 *^^^	0.845 ^^^
	F3	0.86 ± 0.19	0.36 ± 0.052	0.37 ± 0.089	<0.0001 *^^^	<0.0001 *^^^	>0.999 ^^^
	**P4-value**	0.154	<0.0001 *^^^	<0.0001 *^^^			

F1: follow-up 1, F2: follow-up 2, F3: follow-up 3. MDA: malondialdehyde. Statistical tests: ^: two-way ANOVA and Tukey post-hoc test. * Statistically significant (*p*-value < 0.05). P1-value: acetylcysteine vs. control; P2-value: acetylcysteine and Vitamin E vs. control; P3-value: acetylcysteine and Vitamin E vs. acetylcysteine; P4-value: F3 vs. baseline.

**Table 8 biomedicines-13-01275-t008:** Psoriasis indices’ assessments at baseline and at the end of the study among the studied groups.

	PASI Score (Mean ± SD)				DLQI (Mean ± SD)		
	Baseline	F3	% Improvement	P4-Value	Baseline	F3	P4-Value
**Control**	3.47 ± 0.64	3.80 ± 2.21	−6%	0.966	5.53 ± 3.40	6.67 ± 4.50	0.833
**Acetylcysteine**	3.40 ± 0.63	2.07 ± 0.80	42%	0.0217 *	5.93 ± 3.31	2.93 ± 2.90	0.025 *
**Acetylcysteine and Vitamin E**	2.87 ± 0.92	1.40 ± 0.74	52%	0.0083 *	6.20 ± 2.62	2.60 ± 1.50	0.0037 *
P1-value	>0.999	0.0010 *			0.959	0.0182 *	
P2-value	0.697	<0.0001 *			0.913	0.0278 *	
P3-value	0.791	0.594			>0.999	>0.999	

F3: follow-up 3. PASI: Psoriasis Area and Severity Index; DLQI: Dermatology Life Quality Index. Statistical tests: two-way ANOVA and Tukey post-hoc test. * Statistically significant (*p*-value < 0.05). P1-value: acetylcysteine vs. control; P2-value: acetylcysteine and Vitamin E vs. control; P3-value: acetylcysteine and Vitamin E vs. acetylcysteine; P4-value: F3 vs. baseline.

**Table 9 biomedicines-13-01275-t009:** Correlations of IL-36γ and MDA with the PASI score.

		IL-36γ		MDA	
		r	*p*-Value	r	*p*-Value
**Control**	Baseline	0.562	0.037 *	0.751	0.0013 *
	F3	0.655	0.011 *	0.791	0.0004 *
**Acetylcysteine**	Baseline	0.674	0.0059 *	0.756	0.0011 *
	F3	0.640	0.034 *	0.817	0.0012 *
**Acetylcysteine and Vitamin E**	Baseline	0.755	0.011 *	0.753	0.0012 *
	F3	0.715	0.0040 *	0.519	0.040 *

* Statistically significant (*p*-value < 0.05).

## Data Availability

Data is contained in the paper.

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
