# Peer review of "A Promising Approach to Psoriasis Vulgaris Management with N-Acetylcysteine and Vitamin E: Targeting the Interplay of Inflammatory and Oxidative Stress"

_biomedicines, 2025, doi:10.3390/biomedicines13061275_

Round 1

Reviewer 1 Report

Comments and Suggestions for Authors
  1. the routine assays that are reported are not sensitive to 2 decimal places.  This should be corrected throughout the manuscript.
  2. do the authors have any information as to whether the psoriasis is exacerbated once the therapy is stopped?  This often happens with topical steroids.
  3. For completeness the authors may also want to note that both topical and oral calcitriol have been shown to be effective for treating psoriasis.  Unlike calcipotriene topical calcitriol does not cause facial inflammation.

Author Response

Comment 1: the routine assays that are reported are not sensitive to 2 decimal places. This should be corrected throughout the manuscript.

Response 1: Thank you for pointing this out. We agree with this comment. Therefore, the manuscript was revised and corrected and highlighted in yellow.         

Comment 2: do the authors have any information as to whether the psoriasis is exacerbated once the therapy is stopped?  This often happens with topical steroids.

Response 2: Thanks for this valuable inquiry raised by the reviewer. In the current study, NAC and Vitamin E are not considered a replacement for the standard medication, rather, it is an adjunct therapy to improve disease outcomes. Where all the studied groups received their standard medication without replacement during the study course and post-study.

The point in trying NAC and Vitamin E is to assess the added value of consuming antioxidants and anti-inflammatories on the inflammatory module occurring with the psoriasis within the flare-up, remission, and then the stability phase (Tenório et al., 2021)(Wallert et al., 2021).

Usually, the course of the steroids the patient takes during the flare-up is variable, between 15 - 20 days (about 2 to 3 weeks) (Castela et al., 2012). So, what happened during our study is that the patients consumed NAC for 8 weeks only, but our assessment was continued for 1 more month, free from the add-on NAC or Vitamin E.

One main aim of our study, that the add-on therapy (NAC alone or in combination of Vitamin E) can prolong the remission phase for a longer time at the cheapest cost. For example, if the patient used to have 4 or 5 flares annually, with our Add-on, it might be 2-3 times annually or even less (Choon et al., 2023). This will not only affect the patient's health positively but also enhance the quality of life.

Moreover, in our study, we did not specifically evaluate the long-term effects or disease recurrence after cessation of N-acetyl cysteine (NAC) and Vitamin E therapy. However, it is unlikely that these compounds would induce the same rebound effect observed with topical steroids. Unlike steroids, NAC and vitamin E alter oxidative stress and inflammatory pathways rather than suppressing the immune system, which may lower the risk of disease flare-ups after stopping the medication.

We acknowledge that this remains an open question and an area for further investigation. Longer follow-up studies are required in the future to determine whether psoriasis symptoms worsen or reappear after ceasing this treatment. We have added this point to the limitation section and highlight the need for additional research in this area.

Added to Limitation section and highlighted in yellow (Page 23, lines 665-673): "While the current study demonstrates promising results with N-acetyl cysteine and Vitamin E in managing psoriasis vulgaris, it is important to note that the long-term effects and potential recurrence of symptoms after discontinuation of therapy were not evaluated. Unlike topical steroids, which are known to cause rebound flares upon withdrawal due to immune suppression, the antioxidant and anti-inflammatory mechanisms of NAC and Vitamin E may carry a lower risk of such adverse effects. Nevertheless, further studies with extended follow-up periods are warranted to investigate the durability of treatment outcomes and the risk of disease recurrence after therapy cessation."

Castela, E., Archier, E., Devaux, S., Gallini, A., Aractingi, S., Cribier, B., Jullien, D., Aubin, F., Bachelez, H., Joly, P., Le Maître, M., Misery, L., Richard, M. A., Paul, C., & Ortonne, J. P. (2012). Topical corticosteroids in plaque psoriasis: A systematic review of efficacy and treatment modalities. Journal of the European Academy of Dermatology and Venereology, 26(SUPPL. 3), 36–46. https://doi.org/10.1111/J.1468-3083.2012.04522.X,

Choon, S. E., Lebwohl, M. G., Turki, H., Zheng, M., Burden, A. D., Li, L., Quaresma, M., Thoma, C., & Bachelez, H. (2023). Clinical Characteristics and Outcomes of Generalized Pustular Psoriasis Flares. Dermatology (Basel, Switzerland), 239(3), 345. https://doi.org/10.1159/000529274

Tenório, M. C. D. S., Graciliano, N. G., Moura, F. A., de Oliveira, A. C. M., & Goulart, M. O. F. (2021). N-acetylcysteine (Nac): Impacts on human health. Antioxidants, 10(6). https://doi.org/10.3390/ANTIOX10060967

Wallert, M., Börmel, L., & Lorkowski, S. (2021). Inflammatory Diseases and Vitamin E—What Do We Know and Where Do We Go? Molecular Nutrition & Food Research, 65(1), 2000097. https://doi.org/10.1002/MNFR.202000097

Comment 3: For completeness the authors may also want to note that both topical and oral calcitriol have been shown to be effective for treating psoriasis.  Unlike calcipotriene topical calcitriol does not cause facial inflammation.

Response 3: We thank the reviewer for this insightful comment. For the completeness of the information, these comments were added and highlighted in yellow in the revised manuscript (Line 116 - 117, Page 3).               

Reviewer 2 Report

Comments and Suggestions for Authors

The authors present a randomized controlled trial investigating the effect of N-acetyl cysteine and Vitamin E supplementation in patients with psoriasis vulgaris. While the study demonstrates statistically significant improvements, there are major methodological flaws and concerns regarding the clinical significance and interpretation of the findings.

  1. The study included only 15 patients per group, which severely limits the statistical power and generalizability of the findings. Psoriasis is a heterogeneous disease, and such a small cohort is insufficient to derive clinically meaningful conclusions.
  2. The study focuses solely on IL-36γ, while the IL-23/Th17 axis (e.g., IL-17A, IL-23) is the primary and widely recognized immunopathological pathway in psoriasis. The choice of a secondary inflammatory marker limits the biological validity of the conclusions.
  3. Based on the baseline PASI scores (2–3) and the provided clinical photographs, the enrolled patients had very mild psoriasis. In such cases, minimal clinical changes can yield statistically significant differences, but the clinical relevance remains questionable. The improvements observed are unlikely to be meaningful in the broader context of psoriasis management.
  4. The outcomes rely predominantly on PASI and DLQI scores, both of which are subjective. The absence of objective, non-invasive skin measurements such as corneometry, transepidermal water loss (TEWL), or erythema index (Mexameter) weakens the scientific robustness of the study.
  5. Without a Vitamin E-only arm, the independent effect of Vitamin E cannot be evaluated. The current design does not support the conclusion that both NAC and Vitamin E represent novel approaches.
  6. The title suggests a novel therapeutic approach, which is not justified given the very mild severity of disease and the preliminary nature of the findings. It is recommended that the title be revised to accurately reflect the study’s scope and limitations.

Author Response

Comment 1: The study included only 15 patients per group, which severely limits the statistical power and generalizability of the findings. Psoriasis is a heterogeneous disease, and such a small cohort is insufficient to derive clinically meaningful conclusions.

Response 1: Thank you for pointing this out. According to our sample size calculation using G power software, detailed in the manuscript (Page 8-9, lines 338-344), a minimum sample size of 14 was required to achieve a large effect size of 0.5 on PASI score, given the heterogeneity and prevalence of the disease. Moreover, in order to confirm and augment our study findings, we recommend future studies with multi-center involvement and longer duration (page 23, Line 666), and we added this sample size as a study limitation to be improved in future studies (page 23, Line 665).   

Comments 2:

The study focuses solely on IL-36γ, while the IL-23/Th17 axis (e.g., IL-17A, IL-23) is the primary and widely recognized immunopathological pathway in psoriasis. The choice of a secondary inflammatory marker limits the biological validity of the conclusions.

Response 2: Thank you so much for the valuable addressed point. As you just stated, IL-23/Th17 axis is widely recognized, mentioned, and also proved that it plays a major role in the pathway of chronic inflammatory diseases (1), but only few studies focused on the role of IL-36γ and itis importance in psoriasis pathogenicity. High levels in IL-36γ can lead to high expression of IL-17 and IL-23, with a noticeable neutrophile infiltration to the keratinocytes (2). So, in our study, we spotlight on IL-36γ, supporting that it is an important key mediator for psoriasis. This would also open the way for further molecular mechanisms elucidation and deeper investigation that could end up in better treatment options helping the patients.     

Comments 3:

Based on the baseline PASI scores (2–3) and the provided clinical photographs, the enrolled patients had very mild psoriasis. In such cases, minimal clinical changes can yield statistically significant differences, but the clinical relevance remains questionable. The improvements observed are unlikely to be meaningful in the broader context of psoriasis management.

Response 3: We thank the reviewer for this comment. Actually, psoriasis is a disease which is variable in nature, according to a lot of factors. The reaction of the patient's skin towards psoriasis is different, according to, for example (age, skin type, food type, affected area, ...) even if the PASI score for all those different patients is the same. Accordingly, even the patients' preferences usually interfere with the type of treatment recommended to ensure continuity ((3).

Moreover, one of the reasons for choosing PASI (2-3) is to ensure that the patients will not consume systemic medication for treatment, so the effect clearly will not overlap with any other systemic medication. One more important reason for choosing NAC or/with Vit E, is to stabilize the case and slow down its progress. Interestingly, some of the control cases that attended with an initial PASI score of (2-3), progressed to a PASI score of 10 or more, with only the standard medication, which provides evidence on the effectiveness of the currently chosen systemic approach of the study.

Comments 4: The outcomes rely predominantly on PASI and DLQI scores, both of which are subjective. The absence of objective, non-invasive skin measurements such as corneometry, transepidermal water loss (TEWL), or erythema index (Mexameter) weakens the scientific robustness of the study.

Response 4: We thank the reviewer for this insightful comment. In this particular study, we chose PASI and DLQI as primary outcome measures because they are widely accepted and standardized tools in psoriasis research and clinical practice. PASI provides a comprehensive assessment of disease severity by evaluating lesion characteristics (e.g., erythema, induration, and scaling) and affected body surface area, while DLQI captures the patient-reported impact of psoriasis on quality of life (4, 5). These measures were selected to align with the goals of our study, which aimed to assess both clinical improvement and patient-centered outcomes. Kindly note that the psoriatic-specific measurements were performed by a specialized physician, and the included outcomes measures (PASI, DLQI), although limiting, were the measures approved and used in the hospital mentioned.

However, we acknowledge the importance of objective, non-invasive skin measurements in providing additional insights into skin barrier function, hydration, and erythema. Unfortunately, due to resource limitations during the study period, we were unable to incorporate tools such as corneometry, TEWL, or Mexameter. We recognize that this is a limitation of our study and have added a discussion point in the manuscript to address this issue and suggest future research directions.

Comments 5: Without a Vitamin E-only arm, the independent effect of Vitamin E cannot be evaluated. The current design does not support the conclusion that both NAC and Vitamin E represent novel approaches.

Response 5: We thank the reviewer for this valuable comment. Several previous studies assessed Vitamin E levels in chronic skin diseases and stated their lower-than-normal levels (6–8). Thus, in the current study, we primarily aimed to promote the endogenous levels of glutathione through administering its precursor NAC, and we added another group to assess the effect of the combination of NAC with Vitamin E as a complementary add-on to support the anti-oxidant activity. Moreover, we added a recommendation for studies using vitamin E for a longer duration, since the severity of vitamin E deficiency may influence the time required for significant results to be observed (9,10)

Comment 6: The title suggests a novel therapeutic approach, which is not justified given the very mild severity of disease and the preliminary nature of the findings. It is recommended that the title be revised to accurately reflect the study’s scope and limitations.

Response 6: Thank you so much for the valuable addressed point. We are proposing an unconventional way to manage psoriasis patients rather than treat them, through the administration of a widely available and economic antioxidants represented mainly as NAC with possible Vitamin E add-on treatments. The title was carefully revised as per your recommendation and changed into:

“A Promising Approach to Psoriasis Vulgaris Management with N-acetyl Cysteine and Vitamin E: Targeting the Interplay of Inflammatory and Oxidative Stress“ 

References:

  1. Iwakura Y, Ishigame H. The IL-23/IL-17 axis in inflammation. Journal of Clinical Investigation [Internet]. 2006 May 1 [cited 2025 Apr 27];116(5):1218. Available from: https://pmc.ncbi.nlm.nih.gov/articles/PMC1451213/
  2. Sachen KL, Arnold Greving CN, Towne JE. Role of IL-36 cytokines in psoriasis and other inflammatory skin conditions. Cytokine [Internet]. 2022 Aug 1 [cited 2025 Apr 27];156:155897. Available from: https://www.sciencedirect.com/science/article/pii/S1043466622001065#s0055
  3. Ding W, Yao M, Wang Y, Wang M, Zhu Y, Li Y, et al. Patient Needs in Psoriasis Treatment and their Influencing Factors: A Nationwide Multicentre Cross-Sectional Study in China. Indian J Dermatol [Internet]. 2023 Sep 1 [cited 2025 Apr 29];68(5):587. Available from: https://pmc.ncbi.nlm.nih.gov/articles/PMC10718237/
  4. Strober B, Tada Y, Mrowietz U, Lebwohl M, Foley P, Langley RG, et al. Bimekizumab maintenance of response through 3 years in patients with moderate-to-severe plaque psoriasis: results from the BE BRIGHT open-label extension trial. British Journal of Dermatology [Internet]. 2023 Jun 1 [cited 2025 Apr 27];188(6):749–59. Available from: https://pubmed.ncbi.nlm.nih.gov/36967713/
  5. Suriano ES, Souza MDM, Kobata CM, Santos FHY, Mimica MJ. Efficacy of an adjuvant Lactobacillus rhamnosus formula in improving skin lesions as assessed by PASI in patients with plaque psoriasis from a university-affiliated, tertiary-referral hospital in São Paulo (Brazil): a parallel, double-blind, randomized clinical trial. Arch Dermatol Res [Internet]. 2023 Aug 1 [cited 2025 Apr 27];315(6):1621–9. Available from: https://pubmed.ncbi.nlm.nih.gov/36757438/
  6. Zhao H, Guo X, Lei Y, Xia W, Cai F, Zhu D, et al. γ-Tocotrienol inhibits T helper 17 cell differentiation via the IL-6/JAK/STAT3 signaling pathway. Mol Immunol [Internet]. 2022 Nov 1 [cited 2025 Apr 27];151:126–33. Available from: https://pubmed.ncbi.nlm.nih.gov/36126500/
  7. Rocha ACL, Bortoletto MC, da Costa AC, Oyafuso LKM, Sanudo A, Tomita LY. Low serum lycopene, and adequate α-tocopherol levels in patients with psoriasis: A cross-sectional study. Nutr Health [Internet]. 2022 Jun 1 [cited 2025 Apr 27];28(2):239–48. Available from: https://pubmed.ncbi.nlm.nih.gov/33960217/
  8. Liu X, Yang G, Luo M, Lan Q, Shi X, Deng H, et al. Serum vitamin E levels and chronic inflammatory skin diseases: A systematic review and meta-analysis. PLoS One [Internet]. 2021 Dec 1 [cited 2025 Apr 27];16(12 December). Available from: https://pubmed.ncbi.nlm.nih.gov/34905558/
  9. Asbaghi O, Sadeghian M, Nazarian B, Sarreshtedari M, Mozaffari-Khosravi H, Maleki V, et al. The effect of vitamin E supplementation on selected inflammatory biomarkers in adults: a systematic review and meta-analysis of randomized clinical trials. Sci Rep [Internet]. 2020 Dec 1 [cited 2025 Apr 29];10(1):1–17. Available from: https://www.nature.com/articles/s41598-020-73741-6
  10. Ming-Zher Chee N, Prasad Sinnanaidu R, Chan WK, Wah-Kheong Chan C, and G. Vitamin E improves serum markers and histology in adults with metabolic dysfunction-associated steatotic liver disease: Systematic review and meta-analysis. J Gastroenterol Hepatol [Internet]. 2024 Dec 1 [cited 2025 Apr 29];39(12):2545–54. Available from: /doi/pdf/10.1111/jgh.16723

Reviewer 3 Report

Comments and Suggestions for Authors

In this manuscript by Kholy and colleagues, the authors report a randomized cliical trial on the effect of oral administration of N-acetyl cysteine (NAC) alone or in combination with Vitamin E in psoriasis patients. In a total of 45 patients recruited, authors segregate them into three groups with 15 patients each based on power calculations. A control group receiving standard topical steroids, or group (NAC) that received oral NAC along with standard therapy and a group that received both NAC and Vit-E (NAC + Vit-E). Patients were given the treatments for 8 weeks with 3 months follow up. Multiple blood parameters, oxidative stress markers and inflammatory cytokines proteins have been evaluated. Oral administration of NAC alone or NAC + Vit-E did not show any significant differences in liver functionality (assessed by ALT and AST enzymes), lipid profiles, blood glucose and hematological parameters. Inflammatory protein, CRP showed lessor levels in NAC and NAC + Vit-E during follow up. Notably, the pro-inflammatory cytokine IL36g and oxidative stress marker Malondialdehyde (MDA) showed significant reduction in both NAC and NAC + Vit-E groups during follow up compared to control group. This is also reflected in decreased overall psoriasis associated severity index (PASI) score and improved dermatology  life quality index (DLQI) with a concomitant reduction in psoriatic lesions in patients. Notably, no significant differences in the above parameters were noted between NAC versus NAC + Vit-E group suggesting NAC alone could rescue severe psoriatic inflammation and oxidative stress. This study is interesting and has a potential for psoriasis therapeutics, especially in light of the limited therapies available for Psoriasis. This can be accepted provided the following minor concerns are addressed.

Comments:

  1. The experimental design is not specifically clear on the oral administration of the formulation and the follow up. Was NAC/NAC + Vit-E administered for 8 weeks after which the follow up study was conducted or was it parallel (3 months follow up is inclusive of the 8 weeks therapy)?
  2. Even though authors show MDA data as an oxidative stress marker, it would be better to support this by measuring the levels of reduced Glutathione (GSH) since NAC primarily functions via induction of GSH.
  3. Figure 2 & 3, and Figure 4 & 5 are redundant practically depicting the same data in two different ways. Authors could just show Figure 3 and 5 along with statistical values.
  4. It is apparent that Vit-E supplementation does not by itself have any anti-psoriatic activity since for several parameters there is no significant difference between NAC vs NAC + Vit-E group. How do the authors know the orally supplemented Vit-E was bioavailable? May be this could be discussed in light of other literature on Vit-E bioavailability.
  5. Line no 59: Please change “antigen that attacks it” to “pathogen that attacks it”.
  6. Similarly, in line no 59, use simple presentence for general statements (“immune system became active”)

Author Response

Comment 1: The experimental design is not specifically clear on the oral administration of the formulation and the follow up. Was NAC/NAC + Vit-E administered for 8 weeks after which the follow up study was conducted or was it parallel (3 months follow up is inclusive of the 8 weeks therapy)?

Response 1: Thank you for pointing this out. We have conducted the follow-up assessments within (in parallel) the 8 weeks of treatment to monitor any biochemical changes during the treatment period. This was mentioned in the manuscript on Line 260 “Throughout a span of 3 months, patient follow-ups were conducted monthly with comprehensive assessments and examinations of the following aspects”. However, to ensure the clarity of the experimental design, we added “Follow-up assessments were conducted in parallel with the assigned course of treatment” to section 2.4. Study subjects and treatment (Lines 215-216, Page 5) and highlighted it in yellow in the revised manuscript.  

Comment 2: Even though authors show MDA data as an oxidative stress marker, it would be better to support this by measuring the levels of reduced Glutathione (GSH) since NAC primarily functions via induction of GSH.

Response 2: We thank the reviewer for the insightful comment. Firstly, in the study conception, NAC was chosen primarily since it is a proven precursor of glutathione inside the body (1,2). Besides, the effect of NAC as a precursor of glutathione became a fact and no longer a point to debate, so we thought to use a more widely used oxidative marker, like the MDA (3). MDA particularly represents the total oxidative load present regardless of origin and the reduction in its level signifies a marked decrease in the oxidative load (4) It is widely used to measure oxidative stress in autoimmune diseases and various health conditions like allergy-related diseases, asthma, cancer, cardiovascular conditions, and even psychiatric disorders (5–9). Additionally, it is a well-established indicator of oxidative damage in psoriasis and provides valuable insights into the extent of oxidative injury in the skin. One more valid reason is the granted availability of the analytical kit for MDA, due to its extensive usage. However, a recommendation was added to estimate the reduced glutathione levels in future studies. 

Comment 3: Figure 2 & 3, and Figures 4 & 5 are redundant practically depicting the same data in two different ways. Authors could just show Figure 3 and 5 along with statistical values.

Response 3: We thank the reviewer for the valuable observation. Figures 2 & 4 were removed from the manuscript, and the figure numbers were revised to be sequential in a new order.                 

Comment 4: It is apparent that Vit-E supplementation does not by itself have any anti-psoriatic activity since for several parameters there is no significant difference between NAC vs NAC + Vit-E group. How do the authors know the orally supplemented Vit-E was bioavailable? May be this could be discussed in light of other literature on Vit-E bioavailability.

Response 4: We thank the reviewer for the valuable comment. Indeed, Vitamin E by itself does not have anti-psoriatic activity, but rather it supports the antioxidant defense system inside the body against various free radicals. The addition of Vitamin E either to the conventional psoriatic therapy or to NAC and the conventional therapy is a recommended strategy for the management of psoriatic patients whose serum levels of Vitamin E were decreased and seem to have an elevated oxidative stress (10–12). Moreover, in our study, although the group receiving the combination therapy (NAC & Vitamin E) did not show significance in various parameters from the NAC group, but grossly, this group showed better skin quality and condition and complete resolution of the lesion (Figure 7, Page 20).

However, the focus on Vitamin E bioavailability is very insightful, although we used a commercially available Vitamin E supplement, yet the bioavailability may be attributed to the heterogeneity of the disease, and the compliance of the patient in taking the supplement with meals, tis comment was added to the manuscript (line 600 - 603, Page 22). Also, a study found that adding topical nano formulation was an effective adjunct to the psoriasis management protocol (13). Thus, we added a recommendation we recommend the study of the effects of topical Nano formulation of vitamin E in the management of psoriatic lesions” and highlighted it in yellow in the revised manuscript (Line 666 - 667, Page 23).  

Comment 5: Line no 59: Please change “antigen that attacks it” to “pathogen that attacks it”.

Response 5: We appreciate the reviewer for his valuable suggestion. The phrase was corrected and highlighted in yellow in the revised manuscript.                 

Comment 6: Similarly, in line no 59, use simple presentence for general statements (“immune system became active”)

Response 6: We thank the reviewer for the valuable comment. The statement was changed into “The damage activates the immune system” and highlighted in yellow in the revised manuscript.              

References:

1.        Raghu G, Berk M, Campochiaro PA, Jaeschke H, Marenzi G, Richeldi L, et al. The Multifaceted Therapeutic Role of N-Acetylcysteine (NAC) in Disorders Characterized by Oxidative Stress. Curr Neuropharmacol [Internet]. 2020 Dec 31 [cited 2025 Apr 27];19(8):1202–24. Available from: https://pubmed.ncbi.nlm.nih.gov/33380301/

2.        Rushworth GF, Megson IL. Existing and potential therapeutic uses for N-acetylcysteine: The need for conversion to intracellular glutathione for antioxidant benefits. Pharmacol Ther [Internet]. 2014 Feb [cited 2025 Apr 27];141(2):150–9. Available from: https://pubmed.ncbi.nlm.nih.gov/24080471/

3.        Maurya RP, Prajapat MK, Singh VP, Roy M, Todi R, Bosak S, et al. Serum Malondialdehyde as a Biomarker of Oxidative Stress in Patients with Primary Ocular Carcinoma: Impact on Response to Chemotherapy. Clinical Ophthalmology [Internet]. 2021 Feb 26 [cited 2025 Apr 27];15:871–9. Available from: https://www.dovepress.com/serum-malondialdehyde-as-a-biomarker-of-oxidative-stress-in-patients-w-peer-reviewed-fulltext-article-OPTH

4.        Zhang J, Yang Z, Zhang S, Xie Z, Han S, Wang L, et al. Investigation of endogenous malondialdehyde through fluorescent probe MDA-6 during oxidative stress. Anal Chim Acta [Internet]. 2020 Jun 15 [cited 2025 Apr 27];1116:9–15. Available from: https://www.sciencedirect.com/science/article/abs/pii/S000326702030427X?via%3Dihub

5.        Khoubnasabjafari M, Ansarin K, Jouyban A. Reliability of malondialdehyde as a biomarker of oxidative stress in psychological disorders. Bioimpacts [Internet]. 2015 [cited 2025 Apr 27];5(3):123. Available from: https://pmc.ncbi.nlm.nih.gov/articles/PMC4597159/

6.        Maurya RP, Prajapat MK, Singh VP, Roy M, Todi R, Bosak S, et al. Serum Malondialdehyde as a Biomarker of Oxidative Stress in Patients with Primary Ocular Carcinoma: Impact on Response to Chemotherapy. Clinical Ophthalmology [Internet]. 2021 Feb 26 [cited 2025 Apr 27];15:871–9. Available from: https://www.dovepress.com/serum-malondialdehyde-as-a-biomarker-of-oxidative-stress-in-patients-w-peer-reviewed-fulltext-article-OPTH

7.        Merino de Paz N, García-González M, Gómez-Bernal F, Quevedo-Abeledo JC, de Vera-González A, López-Mejias R, et al. Relationship between Malondialdehyde Serum Levels and Disease Features in a Full Characterized Series of 284 Patients with Systemic Lupus Erythematosus. Antioxidants [Internet]. 2023 Aug 1 [cited 2025 Apr 27];12(8):1535. Available from: https://www.mdpi.com/2076-3921/12/8/1535/htm

8.        Nabih GA, Sheshtawy NEe, Mikkawy DMEE, Kamel MA. Serum malondialdehyde as a marker of oxidative stress in rheumatoid arthritis. Egyptian Rheumatology and Rehabilitation [Internet]. 2024 Dec 1 [cited 2025 Apr 27];51(1):1–6. Available from: https://erar.springeropen.com/articles/10.1186/s43166-024-00275-4

9.        Cordiano R, Di Gioacchino M, Mangifesta R, Panzera C, Gangemi S, Minciullo PL. Malondialdehyde as a Potential Oxidative Stress Marker for Allergy-Oriented Diseases: An Update. Molecules [Internet]. 2023 Aug 1 [cited 2025 Apr 27];28(16):5979. Available from: https://pmc.ncbi.nlm.nih.gov/articles/PMC10457993/

10.      Kharaeva Z, Gostova E, De Luca C, Raskovic D, Korkina L. Clinical and biochemical effects of coenzyme Q10, vitamin E, and selenium supplementation to psoriasis patients. Nutrition [Internet]. 2009 Mar [cited 2025 Apr 24];25(3):295–302. Available from: https://pubmed.ncbi.nlm.nih.gov/19041224/

11.      Liu X, Yang G, Luo M, Lan Q, Shi X, Deng H, et al. Serum vitamin E levels and chronic inflammatory skin diseases: A systematic review and meta-analysis. PLoS One [Internet]. 2021 Dec 1 [cited 2025 Apr 24];16(12 December). Available from: https://doi.org/10.1371/journal.pone.0261259

12.      Berardesca E, Cameli N. Vitamin E supplementation in inflammatory skin diseases. Dermatol Ther [Internet]. 2021 Nov 1 [cited 2025 Apr 24];34(6). Available from: https://doi.org/10.1111/dth.15160

13.      Motwani K, Gupta V. Nano-Transfersomes of Vitamin-E and Aloe-Vera for The Management of Psoriasis. Journal of Sustainable Materials Processing and Management [Internet]. 2022 Oct 31 [cited 2025 Apr 24];2(2). Available from: https://doi.org/10.30880/jsmpm.2022.02.02.002

Round 2

Reviewer 1 Report

Comments and Suggestions for Authors

The authors have responded well to my suggested revisions

Author Response

We would like to thank the reviewer for accepting our revisions.

Reviewer 2 Report

Comments and Suggestions for Authors

The authors have addressed the initial comments with diligence and have made several meaningful revisions to improve the manuscript. The study investigates the potential role of N-acetyl cysteine (NAC) and Vitamin E as adjunctive strategies in the management of psoriasis vulgaris, utilizing clinical endpoints (PASI, DLQI) and an inflammatory marker (IL-36γ). The revised manuscript is improved, and many responses were satisfactory. However, a few key issues still warrant further clarification and refinement:

  1. Clinical Relevance in Mild Psoriasis

The study population comprised patients with very mild disease (PASI 2–3), which raises concerns about the clinical meaningfulness of the observed improvements. The authors are encouraged to discuss whether the changes in PASI and DLQI meet established minimal clinically important difference (MCID) thresholds and, if possible, report the proportion of patients achieving PASI50 or PASI75 to provide context.

  1. Interpretation of IL-36γ Findings

While IL-36γ is a biologically relevant cytokine, the absence of parallel measurements of core psoriasis-related markers (e.g., IL-17A, IL-23) limits the mechanistic interpretability. The authors should present IL-36γ as a hypothesis-generating marker, and avoid implying a central role without broader cytokine profiling.

  1. Framing the Study as Exploratory

Given the limited sample size, absence of a Vitamin E-only arm, and lack of objective skin barrier measurements, the study’s findings should be consistently framed as exploratory. Emphasizing this positioning will help align reader expectations with the study’s preliminary scope.

Author Response

Comment 1: Clinical Relevance in Mild Psoriasis

The study population comprised patients with very mild disease (PASI 2–3), which raises concerns about the clinical meaningfulness of the observed improvements. The authors are encouraged to discuss whether the changes in PASI and DLQI meet established minimal clinically important difference (MCID) thresholds and, if possible, report the proportion of patients achieving PASI50 or PASI75 to provide context.

Response 1:

We thank the reviewer for this insightful comment regarding the clinical relevance of our findings in patients with mild psoriasis (PASI 2–3). We appreciate the opportunity to clarify the meaningfulness of the improvement observed. Regarding the MCID thresholds, for PASI, an improvement of ≥2 points or ≥40% from baseline was considered clinically meaningful [1]. In our study, the mean change in PASI was 42% for the Acetylcysteine group, and 52% for the Acetylcysteine & Vitamin E group. Moreover, For DLQI, a reduction of 3.3 points from baseline was considered as the MCID threshold [2] In our study, the reduction in DLQI was 3.6 points for the Acetylcysteine group, and 4 points for the Acetylcysteine & Vitamin E group. These changes exceed the established MCID thresholds, suggesting that even in patients with mild disease, the observed improvements are clinically meaningful. The values mentioned were added to the manuscript in the results section and highlighted in yellow (page 18, lines 459-461, 464-465).

As suggested, we calculated the proportion of patients achieving PASI50 and PASI75 as follows:

  • PASI50: 63.33% of patients achieved at least a 50% improvement in PASI score.
  • PASI75: 23.33% of patients achieved at least a 75% improvement in PASI score.

Although the baseline PASI scores were low, these proportions demonstrate that a meaningful subset of patients experienced substantial improvements in disease severity. This data has been added to the Results section (page 18, lines 461-462).

Comments 2: Interpretation of IL-36γ Findings

While IL-36γ is a biologically relevant cytokine, the absence of parallel measurements of core psoriasis-related markers (e.g., IL-17A, IL-23) limits the mechanistic interpretability. The authors should present IL-36γ as a hypothesis-generating marker, and avoid implying a central role without broader cytokine profiling.

Response 2:

We thank the reviewer for his insightful comment; the primary objective of our study was to explore the potential of use of acetylcysteine as an adjunct therapy for psoriasis. We measured IL-36γ as a type of sensitive inflammatory marker that as been linked to psoriasis in the past 10 years besides being the precursor of other psoriasis-related markers as literature suggests [3–11]

Comments 3: Framing the Study as Exploratory

Given the limited sample size, absence of a Vitamin E-only arm, and lack of objective skin barrier measurements, the study’s findings should be consistently framed as exploratory. Emphasizing this positioning will help align reader expectations with the study’s preliminary scope.

Response 3:

We sincerely thank the reviewer for his constructive feedback which helps ensure that the findings are appropriately contextualized. We have revised the manuscript to consistently frame the study’s findings as exploratory. Besides, throughout the manuscript, we have softened language that might imply definitive conclusions. Also, we have added a limitation statement highlighting the need for future research on larger sample sizes and on multiple centers.

References:

  1. Chow, C.; Simpson, M.J.; Luger, T.A.; Chubb, H.; Ellis, C.N. Comparison of Three Methods for Measuring Psoriasis Severity in Clinical Studies (Part 1 of 2): Change during Therapy in Psoriasis Area and Severity Index, Static Physician’s Global Assessment and Lattice System Physician’s Global Assessment. Journal of the European Academy of Dermatology and Venereology 2015, 29, 1406–1414, doi:10.1111/JDV.13132,.
  2. Basra, M.K.A.; Salek, M.S.; Camilleri, L.; Sturkey, R.; Finlay, A.Y. Determining the Minimal Clinically Important Difference and Responsiveness of the Dermatology Life Quality Index (DLQI): Further Data. Dermatology 2015, 230, 27–33, doi:10.1159/000365390,.
  3. Sachen, K.; Greving, C.; Cytokine, J.T.-; 2022, undefined Role of IL-36 Cytokines in Psoriasis and Other Inflammatory Skin Conditions. Elsevier.
  4. Todorović, V.; Su, Z.; Brent Putman, C.; Kakavas, J.; Salte, K.M.; Mcdonald, H.A.; Wetter, J.B.; Paulsboe, S.E.; Sun, Q.; Gerstein, C.E.; et al. Small Molecule IL-36γ Antagonist as a Novel Therapeutic Approach for Plaque Psoriasis. nature.com, doi:10.1038/s41598-019-45626-w.
  5. Furue, K.; Yamamura, K.; … G.T.-A. dermato; 2018, undefined Highlighting Interleukin-36 Signalling in Plaque Psoriasis and Pustular Psoriasis. pdfs.semanticscholar.org 2018, doi:10.2340/00015555-2808.
  6. Balato, A.; Mattii, M.; Caiazzo, G.; … A.R.-J. of I.; 2016, undefined IL-36γ Is Involved in Psoriasis and Allergic Contact Dermatitis. Elsevier.
  7. Wang, W.; Yu, X.; Wu, C.; medical, H.J.-I. journal of; 2017, undefined IL-36γ Inhibits Differentiation and Induces Inflammation of Keratinocyte via Wnt Signaling Pathway in Psoriasis. pmc.ncbi.nlm.nih.gov.
  8. Chen, J.; Du, J.; Han, Y.; Wei, Z. Correlation Analysis between IL-35, IL-36 γ, CCL27 and Psoriasis Vulgaris. Taylor & Francis 2021, 32, 621–624, doi:10.1080/09546634.2019.1689226.
  9. Guo, J.; Tu, J.; Hu, Y.; Song, G.; Yin, Z. Cathepsin G Cleaves and Activates IL-36γ and Promotes the Inflammation of Psoriasis. Taylor & Francis 2019, 13, 581–588, doi:10.2147/DDDT.S194765.
  10. D’erme, A.; Wilsmann-Theis, D.; … J.W.-J. of I.; 2015, undefined IL-36γ (IL-1F9) Is a Biomarker for Psoriasis Skin Lesions. Elsevier.
  11. Bridgewood, C.; Fearnley, G.W.; Berekmeri, A.; Laws, P.; Macleod, T.; Ponnambalam, S.; Stacey, M.; Graham, A.; Wittmann, M. IL-36γ Is a Strong Inducer of IL-23 in Psoriatic Cells and Activates Angiogenesis. frontiersin.org 2018, 9, doi:10.3389/FIMMU.2018.00200/FULL.
